# FGF23 regulates renal sodium handling and blood pressure

Olena Andrukhova[1], Svetlana Slavic[1], Alina Smorodchenko[1], Ute Zeitz[1], Victoria Shalhoub[2], Beate Lanske[3], Elena E Pohl[1] & Reinhold G Erben[1,*]

## Abstract

Fibroblast growth factor-23 (FGF23) is a bone-derived hormone regulating renal phosphate reabsorption and vitamin D synthesis in renal proximal tubules. Here, we show that FGF23 directly regulates the membrane abundance of the $Na^+$:$Cl^-$ co-transporter NCC in distal renal tubules by a signaling mechanism involving the FGF receptor/αKlotho complex, extracellular signal-regulated kinase 1/2 (ERK1/2), serum/glucocorticoid-regulated kinase 1 (SGK1), and with-no lysine kinase-4 (WNK4). Renal sodium ($Na^+$) reabsorption and distal tubular membrane expression of NCC are reduced in mouse models of *Fgf23* and *αKlotho* deficiency. Conversely, gain of FGF23 function by injection of wild-type mice with recombinant FGF23 or by elevated circulating levels of endogenous Fgf23 in *Hyp* mice increases distal tubular $Na^+$ uptake and membrane abundance of NCC, leading to volume expansion, hypertension, and heart hypertrophy in a αKlotho and dietary $Na^+$-dependent fashion. The NCC inhibitor chlorothiazide abrogates FGF23-induced volume expansion and heart hypertrophy. Our findings suggest that FGF23 is a key regulator of renal $Na^+$ reabsorption and plasma volume, and may explain the association of FGF23 with cardiovascular risk in chronic kidney disease patients.

**Keywords** aldosterone; blood pressure; fibroblast growth factor-23; heart hypertrophy; sodium homeostasis

**Subject Categories** Cardiovascular System; Urogenital System

## Introduction

Fibroblast growth factor-23 (FGF23) is a bone-derived phosphate- and vitamin D-regulating hormone which is secreted by osteocytes and osteoblasts in response to vitamin D and increased extracellular phosphate (The ADHR Consortium, 2000; Saito *et al*, 2005; Martin *et al*, 2012). In the kidney, circulating FGF23 reduces phosphate reabsorption from urine through a direct downregulation of sodium phosphate co-transporters in renal proximal tubular epithelial cells (Shimada *et al*, 2004a,b, 2005; Andrukhova *et al*, 2012). In addition, FGF23 suppresses renal 1α-hydroxylase expression, the key enzyme in vitamin D activation, in proximal tubules (Shimada *et al*, 2001, 2004a). At physiological concentrations, binding of FGF23 to target cells requires co-expression of the ubiquitously expressed FGF receptor-1c and of αKlotho (Urakawa *et al*, 2006), hereafter referred to as Klotho. Klotho is a single-pass transmembrane protein which is mainly expressed in the kidney in renal proximal and distal convoluted tubules, in parathyroid glands, but also in other tissues such as the brain choroid plexus (Kuro-o *et al*, 1997; Hu *et al*, 2010; Andrukhova *et al*, 2012).

In chronic kidney disease (CKD), the declining glomerular filtration rate leads to decreased renal phosphate excretion and subsequent hyperphosphatemia. Hyperphosphatemia in turn stimulates FGF23 secretion from the skeleton. Therefore, FGF23 serum levels increase with CKD progression (Weber *et al*, 2003). It is thought that increased circulating FGF23 helps to maximize renal phosphate excretion during the early stages of CKD (Juppner *et al*, 2010). However, prospective and cross-sectional clinical studies have shown that circulating FGF23 is positively and dose dependently associated with CKD progression, cardiovascular risk factors such as left ventricular hypertrophy, vascular calcifications, and mortality in CKD patients (Juppner *et al*, 2010; Faul *et al*, 2011), suggesting that FGF23 may have additional biological functions which cannot be explained by the known effects of FGF23 on mineral metabolism. The recent report by Faul and coworkers (Faul *et al*, 2011) suggested that FGF23 may induce left ventricular hypertrophy by a direct, Klotho-independent action on cardiomyocytes. In contrast, Xie and coworkers (Xie *et al*, 2012) reported that Klotho may be cardioprotective by an FGF23-independent downregulation of stress-induced calcium channels.

We recently discovered that FGF23 signaling in distal renal tubules upregulates membrane expression of the epithelial calcium channel transient receptor potential vanilloid-5 (TRPV5) by a Klotho-dependent signaling cascade involving extracellular signal-regulated kinase 1 and 2 (ERK1/2), serum/glucocorticoid-regulated kinase 1 (SGK1), and with-no lysine kinase-4 (WNK4) (Andrukhova *et al*, 2014). Both SGK1 and WNK4 are well known to be also

1  University of Veterinary Medicine Vienna, Vienna, Austria
2  Amgen Inc., Thousand Oaks, CA, USA
3  Harvard School of Dental Medicine, Boston, MA, USA
   *Corresponding author. Tel: +43 1 250 77 4550; Fax: +43 1 250 77 4599; E-mail: Reinhold.Erben@vetmeduni.ac.at

involved in renal sodium ($Na^+$) handling. The $Na^+$ and volume-conserving hormone aldosterone increases SGK1 expression and activity, leading to increased renal tubular $Na^+$ reabsorption through augmented membrane abundance of the epithelial $Na^+$ channel (ENaC) in the distal parts of the nephron (Chen *et al*, 1999). Aldosterone is secreted from the adrenal cortex in response to lowered serum $Na^+$, increased serum potassium, and increased circulating angiotensin II. ENaC is a heteromultimeric membrane protein consisting of α-, β-, and γ-subunits. The abundance of the aldosterone-induced α-subunit is the rate-limiting factor in the assembly of the ENaC complex (May *et al*, 1997), whereas the β- and γ-subunits are involved in ubiquitination and degradation of ENaC (Lee *et al*, 2009). WNK4 is an important regulator of distal tubular membrane abundance of the $Na^+$:$Cl^-$ co-transporter NCC and physically interacts with the NCC protein to regulate its membrane trafficking (Cai *et al*, 2006). Patients with WNK4 mutations leading to excessive NCC expression in distal tubules suffer from volume expansion and hypertension (Wilson *et al*, 2001; Kahle *et al*, 2003). Because FGF23 signaling leads to increased serine phosphorylation and activation of SGK1 and WNK4 (Andrukhova *et al*, 2014), we hypothesized that FGF23 may not only regulate the membrane abundance of TRPV5 but also of ENaC and NCC in distal renal tubules. NCC and ENaC are the two key ion channels responsible for $Na^+$ reabsorption in the distal nephron.

# Results

### Fgf23- and Klotho-deficient mice show renal Na+ wasting and are hypovolemic

To test our hypothesis, we first examined $Na^+$ homeostasis in loss-of-function models. Because our earlier studies (Hesse *et al*, 2007; Anour *et al*, 2012; Andrukhova *et al*, 2014) suggested that more subtle effects of *Fgf23* or *Klotho* deficiency on mineral homeostasis might be masked by rapid growth in young mice, we first examined renal $Na^+$ excretion in a non-growing, 9-month-old, compound mutant mouse model characterized by combined loss of *Fgf23* or *Klotho (Kl)* and of a functional vitamin D receptor (VDR). Ablation of *Fgf23* or *Klotho* gene function in mice is associated with early lethality due to uncontrolled production of the active vitamin D hormone and subsequent vitamin D intoxication. However, parallel genetic ablation of vitamin D signaling rescues *Fgf23*$^{-/-}$ and *Kl*$^{-/-}$ mice (Hesse *et al*, 2007; Anour *et al*, 2012), so that *Fgf23*$^{-/-}$/VDR$^{Δ/Δ}$ and *Kl*$^{-/-}$/VDR$^{Δ/Δ}$ double mutant mice can be examined at older ages (Streicher *et al*, 2012). To prevent hypocalcemia and severe hyperparathyroidism in mice with a non-functioning VDR, all mice were kept life-long on a so-called rescue diet rich in calcium, phosphorus, and lactose (Li *et al*, 1998; Erben *et al*, 2002).

Interestingly, both *Fgf23*$^{-/-}$/VDR$^{Δ/Δ}$ and *Kl*$^{-/-}$/VDR$^{Δ/Δ}$ double mutant mice showed renal $Na^+$ wasting relative to VDR mutants and wild-type mice (Fig 1A), which was associated with elevated urinary aldosterone concentrations (Fig 1B). Serum aldosterone levels were higher in *Fgf23*$^{-/-}$/VDR$^{Δ/Δ}$ but not *Kl*$^{-/-}$/VDR$^{Δ/Δ}$ mice relative to wild-type and VDR$^{Δ/Δ}$ mice (Fig 1B). Urinary aldosterone excretion reflects the changes in serum aldosterone over the whole urine sampling period (12 h in our case) and is therefore often more sensitive than serum aldosterone concentration, which reflects only

a specific time point. *Fgf23*$^{-/-}$/VDR$^{Δ/Δ}$ and *Kl*$^{-/-}$/VDR$^{Δ/Δ}$ double mutant mice showed decreased NCC but upregulated membrane abundance of the α-subunit of ENaC relative to wild-type and single VDR mutants as evidenced by immunoblotting of renal membrane preparations and immunohistochemistry (Fig 1C–D). In contrast, the membrane expression of the β- and γ-subunits of ENaC was lower in *Fgf23*$^{-/-}$/VDR$^{Δ/Δ}$ and *Kl*$^{-/-}$/VDR$^{Δ/Δ}$ double mutant mice compared with single VDR mutants (Supplementary Fig S1A). We quantified only the full-length isoforms of α-, β-, and γ-ENAC.

Serum $Na^+$ was not significantly different between *Fgf23*$^{-/-}$/VDR$^{Δ/Δ}$ and *Kl*$^{-/-}$/VDR$^{Δ/Δ}$ compound mutants and VDR$^{Δ/Δ}$ mice (Supplementary Fig S1B). Urinary volume tended to be higher in *Fgf23*$^{-/-}$/VDR$^{Δ/Δ}$ and *Kl*$^{-/-}$/VDR$^{Δ/Δ}$ double mutant mice relative to wild-type mice, but was not significantly changed relative to VDR$^{Δ/Δ}$ mice (Supplementary Fig S1B). Serum potassium remained unchanged in *Kl*$^{-/-}$/VDR$^{Δ/Δ}$ and was actually lower in *Fgf23*$^{-/-}$/VDR$^{Δ/Δ}$ compared with wild-type and VDR$^{Δ/Δ}$ mice (Supplementary Fig S1B), ruling out hyperkalemia as the driving force for increased aldosterone secretion in compound mutant mice. Urinary potassium excretion, urinary volume, and urinary pH did not differ between VDR$^{Δ/Δ}$ and *Kl*$^{-/-}$/VDR$^{Δ/Δ}$ or *Fgf23*$^{-/-}$/VDR$^{Δ/Δ}$ mice (Supplementary Fig S1B). To rule out differences in dietary $Na^+$ intake as a possible cause of increased urinary $Na^+$ excretion in *Kl*$^{-/-}$/VDR$^{Δ/Δ}$ and *Fgf23*$^{-/-}$/VDR$^{Δ/Δ}$ mice, we measured food consumption over 1 week. However, mean food consumption, and thus $Na^+$ intake, did not differ between the groups (Supplementary Fig S1C). Rather, our data suggest that the *Fgf23* and *Klotho* deficiency-induced down-regulation of NCC causes increased urinary $Na^+$ excretion, overriding the counter-regulatory and probably aldosterone-driven increase in α-ENaC expression.

To examine whether similar changes would be present in *Fgf23*$^{-/-}$ and *Kl*$^{-/-}$ mice, we examined $Na^+$ homeostasis at 4 weeks of age, when *Fgf23*$^{-/-}$ and *Kl*$^{-/-}$ mice are still viable. Although renal $Na^+$ wasting was observed only in *Fgf23*$^{-/-}$/VDR$^{Δ/Δ}$ mice, 4-week-old *Fgf23*$^{-/-}$, *Kl*$^{-/-}$, *Fgf23*$^{-/-}$/VDR$^{Δ/Δ}$, and *Kl*$^{-/-}$/VDR$^{Δ/Δ}$ mice displayed downregulated renal NCC expression, upregulated urinary aldosterone, and increased renal α-ENaC expression (Supplementary Fig S2), in very good agreement with the data from 9-month-old mice. Consistent with our findings, increased serum aldosterone in hypomorphic *Kl/Kl* mice was previously reported also by other investigators (Fischer *et al*, 2010). The phenotypes of the originally described hypomorphic *Kl/Kl* mouse (Kuro-o *et al*, 1997) and of Klotho null mice (*Kl*$^{-/-}$) generated later are identical (Tsujikawa *et al*, 2003).

Collectively, these results demonstrate that *Fgf23* and *Klotho* deficiency leads to decreased membrane expression of NCC in renal distal tubules in young and aged mice, and subsequently to renal $Na^+$ wasting in non-growing mice despite elevated aldosterone secretion. It is well known that activity of the NCC channel is regulated by protein phosphorylation at different sites (Pacheco-Alvarez *et al*, 2006). Therefore, we assessed NCC phosphorylation at serine 71 and 91 and threonine 55 by immunoblotting. As shown in Supplementary Fig S1D, abundance of phospho-NCC was reduced in 9-month-old *Fgf23*$^{-/-}$/VDR$^{Δ/Δ}$ and *Kl*$^{-/-}$/VDR$^{Δ/Δ}$ mutants relative to wild-type and single VDR mutant mice. These findings are in accordance with the notion that loss of *Fgf23* and *Klotho* function leads to decreased membrane transport and activation of the NCC channel.

It is clear that despite chronically increased urinary $Na^+$ loss, 9-month-old *Fgf23*$^{-/-}$/VDR$^{Δ/Δ}$ and *Kl*$^{-/-}$/VDR$^{Δ/Δ}$ mice must be in a

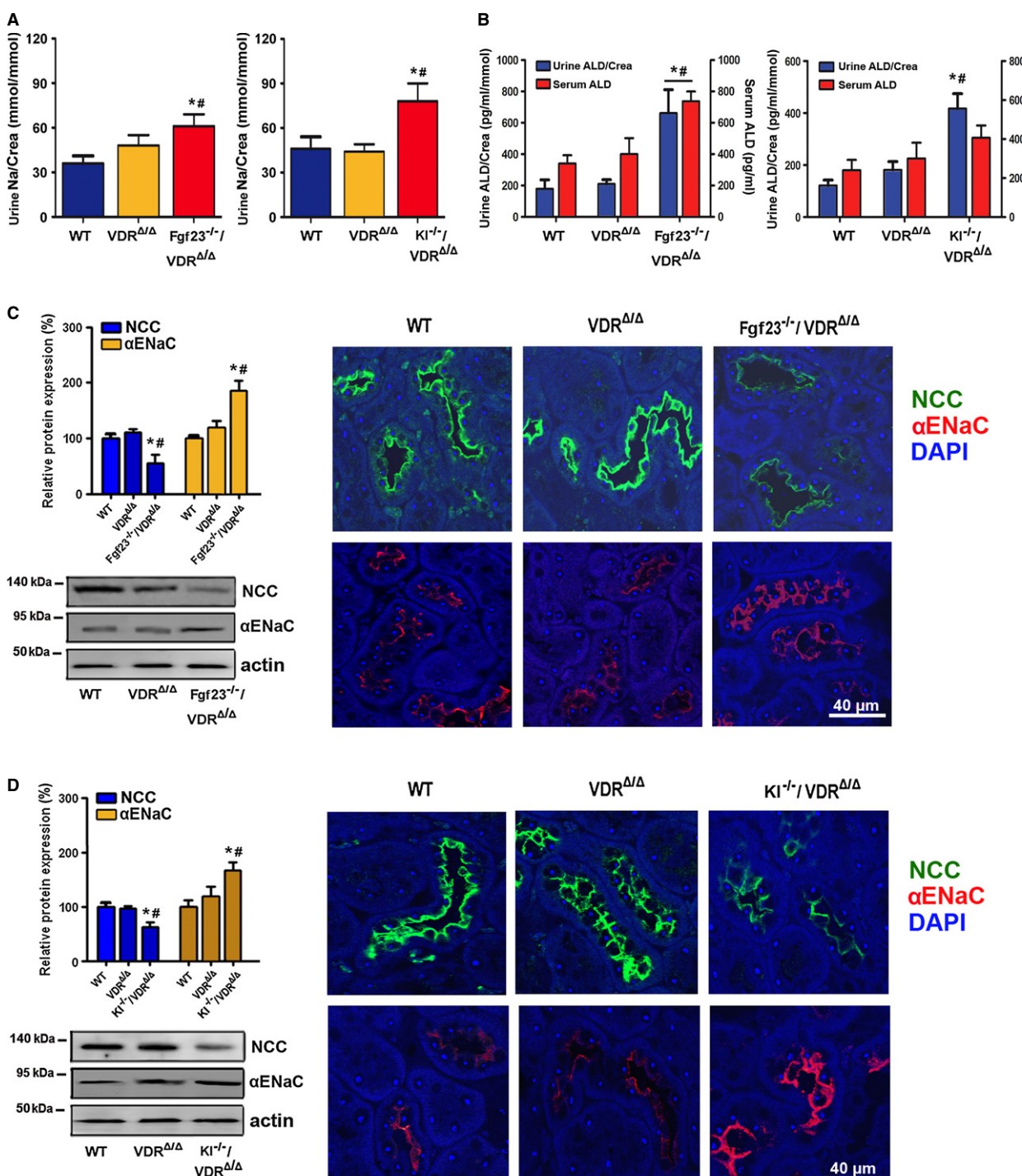

**Figure 1. *Fgf23* or *Klotho* deficiency induces renal sodium wasting caused by reduced expression of the Na⁺:Cl⁻ co-transporter NCC.**

A, B   (A) Urinary Na⁺ excretion corrected by urinary creatinine (Crea) (*n* = 10–12, one-way ANOVA followed by SNK test, \*$P$ = 0.0114 versus WT, #$P$ = 0.0325 versus VDR$^{Δ/Δ}$ for *Fgf23*$^{−/−}$/VDR$^{Δ/Δ}$, \*$P$ = 0.0218 versus WT, #$P$ = 0.0185 versus VDR$^{Δ/Δ}$ for *Kl*$^{−/−}$/VDR$^{Δ/Δ}$) and (B) urinary aldosterone concentration corrected by urinary creatinine and serum aldosterone concentration measured by ELISA (*n* = 8–10, one-way ANOVA followed by SNK test, \*$P$ < 0.05 versus WT, #$P$ < 0.05 versus VDR$^{Δ/Δ}$), in 9-month-old male wild-type (WT), VDR$^{Δ/Δ}$, *Fgf23*$^{−/−}$/VDR$^{Δ/Δ}$, or *Kl*$^{−/−}$/VDR$^{Δ/Δ}$ compound mutant mice on the rescue diet.

C, D   Western blotting analysis of NCC and α-ENaC protein expression in renal cortical total membrane fractions (*n* = 7–9, one-way ANOVA followed by SNK test, \*$P$ < 0.005 versus WT, #$P$ < 0.005 versus VDR$^{Δ/Δ}$), and immunohistochemical detection of NCC and α-ENaC protein expression in paraffin sections of paraformaldehyde-fixed kidneys (*n* = 3–5) in 9-month-old male wild-type (WT), VDR$^{Δ/Δ}$, *Fgf23*$^{−/−}$/VDR$^{Δ/Δ}$, or *Kl*$^{−/−}$/VDR$^{Δ/Δ}$ compound mutant mice on the rescue diet. Data represent mean ± s.e.m.

Source data are available for this figure.

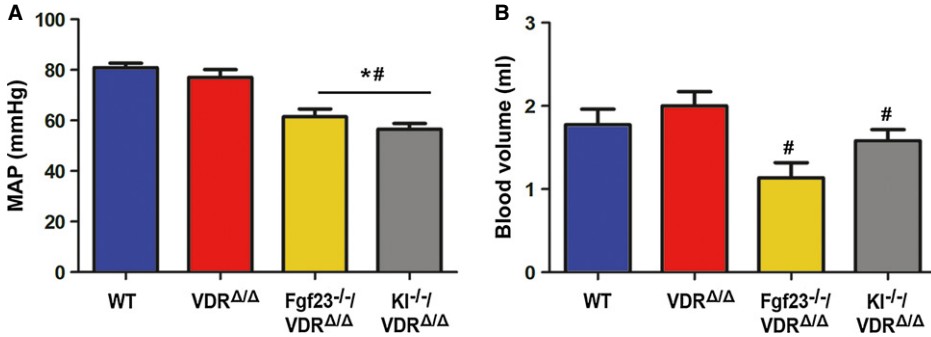

**Figure 2. Decreased mean arterial pressure and blood volume in *Fgf23*- or *Klotho*-deficient mice.**

A, B Mean arterial pressure (A) and blood volume (B) ($n = 6–8$, one-way ANOVA followed by SNK test, $*P < 0.05$ versus WT, $^{\#}P < 0.05$ versus VDR$^{\Delta/\Delta}$) in 9-month-old male wild-type (WT), VDR$^{\Delta/\Delta}$, *Fgf23*$^{-/-}$/VDR$^{\Delta/\Delta}$, or *Kl*$^{-/-}$/VDR$^{\Delta/\Delta}$ compound mutant mice on the rescue diet. Data represent mean $\pm$ s.e.m.

steady state. Our data suggest that in order to adapt to reduced NCC expression/activation and the accompanying renal Na$^+$ wasting, *Fgf23*$^{-/-}$/VDR$^{\Delta/\Delta}$ and *Kl*$^{-/-}$/VDR$^{\Delta/\Delta}$ mutants upregulate aldosterone to conserve Na$^+$ in aldosterone target organs and to maintain normal serum Na$^+$ and osmolarity. Therefore, although food intake was not different between the genotypes, it is likely that compound mutants had higher intestinal Na$^+$ absorption due to increased aldosterone. In this explanatory model, the driving force behind increased aldosterone secretion in compound mutants would be hypovolemia, leading to activation of the renin-angiotensin-aldosterone system. Therefore, we assessed blood volume and blood pressure in 9-month-old *Fgf23*$^{-/-}$/VDR$^{\Delta/\Delta}$ and *Kl*$^{-/-}$/VDR$^{\Delta/\Delta}$ mice. Indeed, we found lower blood pressure and volume in *Fgf23*$^{-/-}$/VDR$^{\Delta/\Delta}$ and *Kl*$^{-/-}$/VDR$^{\Delta/\Delta}$ relative to VDR$^{\Delta/\Delta}$ mice (Fig 2). However, we were unable to detect differences in plasma renin activity between the genotypes, using a commercial assay (Supplementary Fig S1E). We don't have a good explanation why plasma renin activity remained unchanged in *Fgf23*$^{-/-}$/VDR$^{\Delta/\Delta}$ and *Kl*$^{-/-}$/VDR$^{\Delta/\Delta}$ mice despite hypovolemia. Because the observed increases in urinary aldosterone excretion in *Fgf23*$^{-/-}$/VDR$^{\Delta/\Delta}$ and *Kl*$^{-/-}$/VDR$^{\Delta/\Delta}$ mutants were mild, it is possible that the changes in plasma renin activity were too small to be picked up by the assay. Taken together, serum Na$^+$ concentrations are maintained in *Fgf23*$^{-/-}$/VDR$^{\Delta/\Delta}$ and *Kl*$^{-/-}$/VDR$^{\Delta/\Delta}$ mutants at the expense of reduced blood volume and hypotension.

**Recombinant FGF23 directly upregulates distal tubular NCC and causes hypertension**

Next, we examined gain-of-function models. As expected, intraperitoneal injection of 10 μg recombinant FGF23 (rFGF23) over 5 days into 3-month-old wild-type mice caused hyperphosphaturia and hypophosphatemia (Supplementary Fig S3). In addition, rFGF23 profoundly reduced urine volume, reduced renal Na$^+$ excretion, and increased blood Na$^+$ concentration (Fig 3A). Serum and urinary aldosterone was suppressed in rFGF23-treated mice, whereas plasma renin activity remained unchanged (Fig 3A). NCC was about 40% upregulated in renal membrane preparations from rFGF23-treated wild-type mice relative to vehicle controls (Fig 3B). Immunohistochemistry also showed increased NCC staining of the luminal cell membranes in distal tubules after rFGF23 treatment

(Fig 3C). In addition, rFGF23 treatment increased the abundance of phosphorylated NCC at serine 71 and 91 in renal membrane preparations (Fig 3D). After correction for total NCC expression, the strongest effect of rFGF23 on NCC phosphorylation was observed at serine 71 (Supplementary Fig S4). Inversely to our findings in loss-of-function models, serum and urinary aldosterone as well as renal α-ENaC expression was downregulated, whereas renal expression of the full-length β- and γ-subunits of ENaC was increased by rFGF23 (Fig 3A–C). In agreement with the notion that WNK4 physically interacts with the NCC protein to regulate its membrane trafficking (Cai *et al*, 2006) and that FGF23 signaling activates WNK4 (Andrukhova *et al*, 2014), we found higher WNK4 serine phosphorylation and an increased association between WNK4 and NCC in kidney homogenates of rFGF23-treated mice (Fig 3E). We confirmed the specificity of the anti-NCC and anti-WNK4 antibodies by using extracts from kidneys of NCC- and WNK4-knockout mice, respectively (Supplementary Fig S5).

To examine whether the rFGF23-induced upregulation in membrane expression and phosphorylation of NCC is associated with increased Na$^+$ uptake in distal tubular epithelium, we performed intracellular Na$^+$ imaging in live kidney slices, using 2-photon microscopy. Three-hundred-μm-thick kidney slices were prepared from wild-type mice treated with vehicle or rFGF23 8 h before necropsy. The slices were stained with the fluorescent intravital Na$^+$ indicator SFBI (Harootunian *et al*, 1989), which was excited at a wavelength of 820 nm. Fig 3F shows that rFGF23 treatment induced an about threefold increase in fluorescence intensity in distal tubules. SFBI fluorescence intensity in distal tubules of rFGF23-treated mice returned to normal within 30 min after *ex vivo* addition of the thiazide diuretic chlorothiazide (Fig 3F), a well-known functional blocker of the NCC channel (Monroy *et al*, 2000). In addition, we treated live SFBI-loaded kidney slices prepared from wild-type mice with rFGF23 or vehicle *in vitro*. rFGF23 gradually increased intracellular SFBI fluorescence over 105 min in distal tubules, whereas fluorescence intensity remained unchanged in vehicle-treated slices (Fig 3G and Supplementary Videos S1 and S2). The rFGF23-induced increase in intracellular Na$^+$ concentration was reversed by chlorothiazide within 30 min (Fig 3G and Supplementary Video S1). Taken together, these data show that rFGF23 increases NCC membrane abundance and phosphorylation and activates Na$^+$ uptake in distal renal tubules *in vivo* and *in vitro*.

As a consequence of renal Na$^+$ retention and volume expansion, a 5-day rFGF23 treatment increased diastolic, systolic, and mean arterial blood pressure by about 20 mm Hg (Fig 4A). The heart/body weight ratio was increased and cross-sections of the heart showed thickening of the ventricular septum in rFGF23-treated mice after only 5 days of treatment (Fig 4A). In contrast to wild-type and VDR$^{\Delta/\Delta}$ mutant mice, rFGF23 treatment of 3-month-old $Kl^{-/-}$/VDR$^{\Delta/\Delta}$ double mutant mice did not result in renal Na$^+$ retention

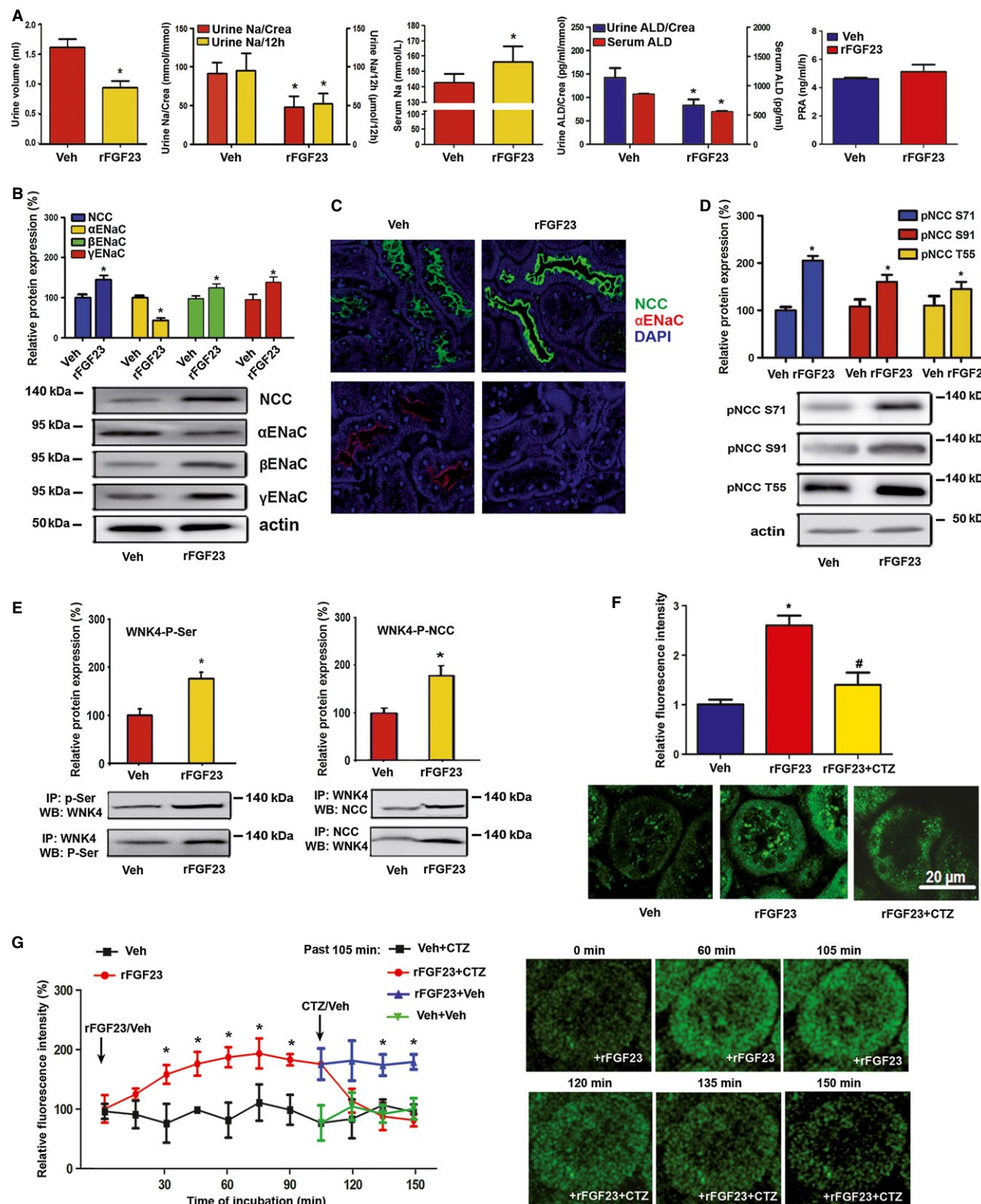

**Figure 3.**

**Figure 3.   Gain of FGF23 function induces renal Na$^+$ retention through increased renal NCC expression and channel activation.**

A   Urine volume ($n$ = 15–17), urinary Na$^+$ excretion per 12 h ($n$ = 15–17, Students $t$-test, *$P$ = 0.0085), urinary Na$^+$ excretion corrected by urinary creatinine (Crea) ($n$ = 15–17, Students $t$-test, *$P$ = 0.0251), serum Na$^+$ concentration ($n$ = 15–17, Students $t$-test, *$P$ = 0.0308), serum and urinary aldosterone concentrations corrected by urinary creatinine ($n$ = 4–5, Students $t$-test, * serum $P$ = 0.0040, urine $P$ = 0.0156), and plasma renin activity (RPA) ($n$ = 4–5) after 5 days of treatment of 3-month-old male wild-type mice with vehicle (Veh) or recombinant FGF23 (10 μg per mouse per day).

B, C   Western blotting quantification (B) of NCC, α-ENaC, β-ENaC, and γ-ENaC protein expression in renal cortical total membrane fractions ($n$ = 4–5, Students $t$-test, *NCC $P$ = 0.0014, α-ENaC $P$ = 0.0007, β-ENaC $P$ = 0.0251, γ-ENaC $P$ = 0.0344), and immunohistochemical detection (C) of NCC and α-ENaC protein expression in kidney sections of 3-month-old wild-type mice treated for 5 days with vehicle or rFGF23 ($n$ = 3–4).

D   Western blotting quantification of NCC phosphorylation at Ser71, Ser91, and Thr58 (pNCC S71, pNCC S91, pNCC T55) in total kidney homogenates of 3-month-old wild-type mice treated for 5 days with vehicle or rFGF23 ($n$ = 4–5, Students $t$-test, *pNCC S71 $P$ = 0.0001, pNCC S91 $P$ = 0.0182, pNCC T55 $P$ = 0.0056).

E   Reciprocal immunoprecipitation (IP) of serine-phosphorylated (P-Ser) proteins, followed by Western blot (WB) analysis of WNK4 or vice versa from homogenized renal cortex protein samples of 3-month-old male wild-type mice treated for 5 days with vehicle or rFGF23 ($n$ = 5–6, Students $t$-test, *WNK4-P-Ser $P$ = 0.0057). For co-immunoprecipitation of NCC/WNK4 complexes, WNK4 or NCC were immunoprecipitated with specific antibodies (anti-NCC and anti-WNK4) from homogenized renal cortex protein samples of 3-month-old wild-type mice treated for 5 days with vehicle or rFGF23. Western blot analysis was performed with corresponding anti-NCC or anti-WNK4 antibodies to identify co-precipitated NCC and WNK4 protein, respectively ($n$ = 4–6, Students $t$-test, WNK4-P-NCC *$P$ = 0.0116).

F   Quantification and original images of intracellular Na$^+$ levels in renal distal tubular cells in live 300-μm-thick kidney slices of 3-month-old WT mice treated with vehicle or rFGF23 (10 μg/mouse), 8 h before necropsy ($n$ = 4, one-way ANOVA followed by SNK test, *$P$ = 0.0026 versus vehicle-treated mice, #$P$ = 0.0175 versus rFGF23-treated mice). Kidney slices were stained with the sodium-sensitive dye SBFI. Chlorothiazide (CTZ, 10 μM) was used as NCC inhibitor.

G   Time-dependent changes in intracellular Na$^+$ levels in renal distal tubules in SBFI-loaded, 300-μm-thick, live kidney slices of 3-month-old WT mice treated *in vitro* at time 0 with rFGF23 (100 ng/ml) or vehicle ($n$ = 3–6). After 105 min, 10 μM CTZ or vehicle was added. Fluorescence intensity in G and H was quantified in 4–9 regions of interest per image, sample, and time point from 2–3 independent experiments. Students $t$-test, *$P$ < 0.05 versus vehicle-treated or versus vehicle- + CTZ vehicle-treated (past 105 min). Data represent mean ± s.e.m.

Source data are available for this figure.

and heart hypertrophy (Fig 4B), showing that the effects of FGF23 on Na$^+$ homeostasis and heart hypertrophy are Klotho dependent, but VDR independent.

It was reported that *Klotho*-deficient mice develop heart hypertrophy caused by increased circulating Fgf23 (Faul *et al*, 2011). However, we actually found a decreased heart/body weight ratio in 4-week-old $Kl^{-/-}$ mice, and unchanged heart/body weight ratio in

$Kl^{-/-}$/VDR$^{\Delta/\Delta}$ mice compared to wild-type and VDR$^{\Delta/\Delta}$ littermates (Supplementary Fig S6), suggesting that chronically elevated endogenous Fgf23 serum levels in $Kl^{-/-}$ mice do not cause heart hypertrophy in the absence of the co-receptor Klotho.

To verify that the regulation of NCC by FGF23 is a direct effect on the distal tubule, we isolated distal tubular segments from wild-type and *Fgf23*-deficient mice and treated these segments with

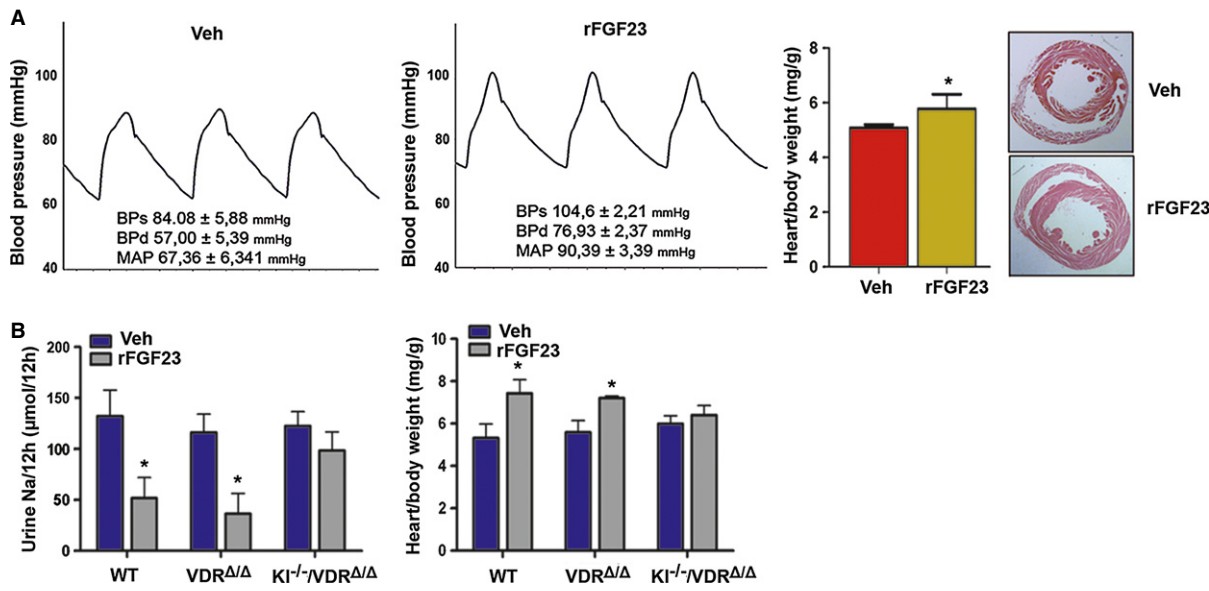

**Figure 4.   FGF23 administration induces hypertension and heart hypertrophy in a Klotho-dependent fashion.**

A   Representative aortic blood pressure curves, heart weight/body weight ratios, and H&E-stained paraffin cross-sections of hearts from 3-month-old male wild-type mice treated for 5 days with vehicle or 10 μg rFGF23 per mouse per day ($n$ = 5–6, Students $t$-test, *$P$ = 0.0216). Insets show systolic (BPs), diastolic (BPd), and mean arterial pressure (MAP).

B   Urinary Na$^+$ excretion per 12 h and heart weight/body weight ratio in 3-month-old male wild-type, VDR$^{\Delta/\Delta}$, and $Kl^{-/-}$/VDR$^{\Delta/\Delta}$ compound mutant mice treated for 5 days with vehicle or 10 μg rFGF23 per mouse per day ($n$ = 4–6, Students $t$-test, *$P$ < 0.05 versus vehicle). Data represent mean ± s.e.m.

rFGF23 alone or in combination with specific ERK1/2 and SGK1 inhibitors for 2 h *in vitro*. rFGF23 upregulated NCC protein expression in distal tubular segments from wild-type and *Fgf23*-deficient mice (Fig 5A). This effect was blocked by ERK1/2 or SGK1 inhibitors (Fig 5A), showing that the ERK1/2-SGK1 signaling pathway is essential for the direct, FGF23-induced regulation of NCC expression in the distal tubule. In addition, rFGF23 increased NCC phosphorylation at serine 71 in distal tubules from wild-type but not from *Klotho*-deficient mice, indicating that the co-receptor Klotho is essential for the FGF23-induced phosphorylation of NCC (Fig 5B). rFGF23 had no effect on ENaC expression in distal tubules isolated from wild-type mice, further supporting the notion that the inhibitory effects of rFGF23 on ENaC expression *in vivo* are indirect effects mediated through the suppression of aldosterone secretion (Fig 5C).

## The NCC inhibitor chlorothiazide abrogates the hypertensive effects of FGF23

Next, we reasoned that if indeed the cardiovascular effects of rFGF23 were mediated by $Na^+$ retention through upregulated distal renal tubular NCC expression, an inhibitor of NCC function should prevent the rFGF23-mediated rise in circulating blood volume, blood pressure, and heart/body weight ratio. As expected, treatment of wild-type mice with the NCC inhibitor chlorothiazide increased urine volume and renal $Na^+$ excretion, but did not change blood volume, central venous pressure, arterial blood pressure, or heart/body weight ratio (Fig 6). However, co-treatment of wild-type mice with rFGF23 and chlorothiazide completely prevented the rFGF23-induced $Na^+$ retention, volume expansion, rise in central venous and arterial blood pressure, heart hypertrophy, and rise in cardiac expression of the hypertrophy-associated gene β-myosin heavy chain (Fig 6 and Supplementary Fig S7). These results clearly indicate that the cardiovascular effects of increased circulating FGF23

are mediated through upregulation of distal renal tubular NCC and consequently higher renal tubular reabsorption of $Na^+$.

## Dietary $Na^+$ modulates the effects of FGF23 on renal $Na^+$ handling and blood pressure

To assess the modulatory effect of dietary $Na^+$ on the hypertensive effect of FGF23, we fed diets with different $Na^+$ content to wild-type mice and treated them for 5 days with vehicle or rFGF23. Analysis of the data by two-way ANOVA showed a significant interaction between the diet and the rFGF23-induced increase in blood pressure (Fig 7A). However, much to our surprise, the rFGF23-induced increase in arterial blood pressure was inversely related to dietary $Na^+$, that is, stronger on the low $Na^+$ diet (Fig 7A). To find an explanation for this puzzling finding, we analyzed serum and urinary $Na^+$ and aldosterone together with renal expression of NCC and of α-, β-, and γ-ENaC subunits. In analogy to the effects on blood pressure, the rFGF23-induced increase in serum $Na^+$ and the suppression of urinary $Na^+$ excretion (in absolute numbers) were most pronounced on the low $Na^+$ diet (Fig 7B). As expected, serum and especially urinary aldosterone were inversely related to dietary $Na^+$ content in vehicle-treated mice (Fig 7C). The 5-day treatment with rFGF23 suppressed serum and urinary aldosterone on the low and normal $Na^+$ diets (Fig 7C). However, the remaining levels of urinary aldosterone excretion in rFGF23-treated mice were inversely related to dietary $Na^+$ content (Fig 7C).

rFGF23-treated mice showed increased renal NCC expression compared with vehicle controls on all three diets, but the level of NCC abundance was profoundly modulated by dietary $Na^+$ (Fig 7D). NCC abundance was more than twofold higher in kidneys of rFGF23-treated mice on low $Na^+$ compared with those on high $Na^+$ diet (Fig 7D). Interestingly, rFGF23 treatment downregulated renal expression of α-ENaC and upregulated expression of the β- and γ-ENaC subunits on the normal and high $Na^+$ diet, but had

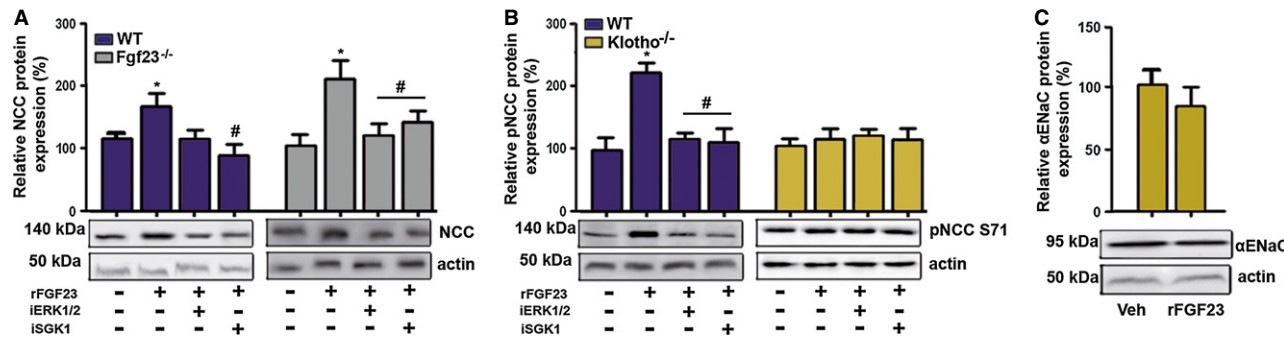

**Figure 5.  FGF23 regulates NCC expression and phosphorylation in isolated distal tubular segments in a Klotho-dependent manner.**

A  Western blotting quantification of NCC expression in isolated distal tubular segments from wild-type (WT) and $Fgf23^{-/-}$ mice treated for 2 h *in vitro* with vehicle or rFGF23 alone or in combination with specific ERK1/2 (iERK1/2) or SGK1 inhibitors (iSGK1) ($n = 4$–6, one-way ANOVA followed by SNK test, *$P < 0.05$ versus vehicle, #$P < 0.05$ versus rFGF23 alone).

B  Western blotting quantification of NCC phosphorylation at Ser71 (pNCC S71) in isolated distal tubular segments from WT and $Kl^{-/-}$ mice treated for 2 h *in vitro* with vehicle or rFGF23 alone or in combination with specific ERK1/2 or SGK1 inhibitors ($n = 4$, one-way ANOVA followed by SNK test, *$P = 0.0002$ versus vehicle, #$P < 0.005$ versus rFGF23 alone).

C  Western blotting quantification of α-ENaC protein expression in isolated distal tubular segments from WT mice treated for 2 h *in vitro* with vehicle (Veh) or rFGF23 ($n = 3$–4). Data represent mean ± s.e.m.

Source data are available for this figure.

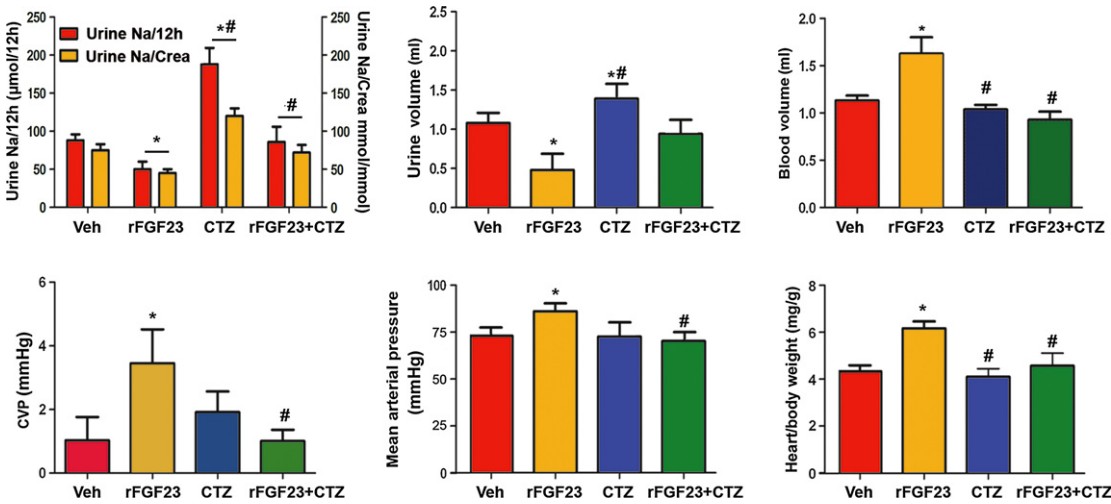

**Figure 6.  Co-treatment of mice with rFGF23 and chlorothiazide abrogates the untoward cardiovascular effects of rFGF23.**
Urinary Na$^+$ excretion per 12 h, urine volume, blood volume, central venous pressure, mean arterial pressure, and heart/body weight ratio in 3-month-old male wild-type mice treated for 5 days with vehicle (Veh), recombinant FGF23 (10 μg per mouse per day), or chlorothiazide (CTZ, 25 mg/kg) alone or in combination ($n = 8$–10). One-way ANOVA followed by SNK test, *$P < 0.05$ versus vehicle, #$P < 0.05$ versus rFGF23. Data represent mean ± s.e.m.

the opposite effect on the low Na$^+$ diet (Fig 7D). These findings are consistent with the notion that FGF23 does not directly regulate ENaC membrane abundance, but that higher residual aldosterone levels on the low Na$^+$ diet interfere with the counter-regulatory suppression of α-ENaC seen in the distal nephron of rFGF23-treated mice on the normal and high Na$^+$ diets. It is well known that aldosterone signaling increases transcription and activation of SGK1 (Chen *et al*, 1999). Therefore, we hypothesized that FGF23 and aldosterone signaling might converge on SGK1, resulting in over-additive effects on SGK1 activation in rFGF23-treated mice on the low Na$^+$ diet. To test this, we analyzed phosphorylated (pSGK1) and total SGK1 in renal homogenates by immunoblotting. We found that the ratio of pSGK1 versus total SGK1 was inversely associated with the dietary Na$^+$ content in rFGF23-treated mice (Fig 7E), corroborating the notion that higher aldosterone levels on the low Na$^+$ diet augmented the rFGF23-induced SGK1 activation. Collectively, these results show that FGF23 and aldosterone signaling pathways interact in the activation of SGK1 and the regulation of Na$^+$ reabsorption in the distal nephron.

**Hyp mice show overexpression of NCC and hypertension**

Finally, we assessed the cardiovascular effects of chronically elevated endogenous Fgf23 in *Hyp* mice, a model of human X-linked hypophosphatemia (XLH). *Hyp* mice and XLH patients are characterized by loss-of-function mutations in PHEX (phosphate-regulating gene with homologies to endopeptidases on the X-chromosome), leading to impaired bone mineralization and subsequently increased biosynthesis of Fgf23 (Liu *et al*, 2003; Barros *et al*, 2013). As expected, *Hyp* mice showed about 20-fold increased serum levels of intact Fgf23 (Fig 8A). In accordance with our findings in rFGF23-treated mice, chronically elevated circulating Fgf23 in *Hyp* mice was associated with increased heart-to-body weight ratio, elevated serum Na$^+$, and decreased urinary Na$^+$ excretion (Fig 8B). Moreover, mean arterial blood pressure (Fig 8C), renal NCC membrane

expression (Fig 8D), and NCC phosphorylation at S71, S91, and T58 (Fig 8D) were increased in *Hyp* mice, relative to wild-type controls. Similar to the findings in rFGF23-treated wild-type mice, serum and urinary aldosterone was suppressed in *Hyp* mice compared with wild-type controls (Fig 8E). Thus, *Hyp* mice recapitulate the changes in Na$^+$ homeostasis and blood pressure found in rFGF23-treated wild-type mice.

## Discussion

Our study suggests that FGF23 directly regulates NCC membrane abundance and activity in distal renal tubules through its canonical signaling pathway involving the FGF receptor 1c/αKlotho-ERK1/2-SGK1-WNK4 signaling axis. Thus, FGF23 is not only a phosphaturic, but also a Na$^+$-conserving hormone involved in volume and blood pressure homeostasis. This new paradigm describing the novel FGF23-mediated bone-kidney-heart axis is shown in Fig 9.

Loss-of-function mutations in NCC result in Gitelman's syndrome in humans (Naesens *et al*, 2004). Gitelman's syndrome is characterized by normal to low blood pressure, hypokalemia, hypocalciuria, and metabolic alkalosis. Although *Fgf23*$^{-/-}$/VDR$^{\Delta/\Delta}$ and *Kl*$^{-/-}$/VDR$^{\Delta/\Delta}$ mutants show reduced NCC expression and increased urinary Na$^+$ excretion, they do not develop a typical Gitelman's syndrome. Rather, *Fgf23*$^{-/-}$/VDR$^{\Delta/\Delta}$ and *Kl*$^{-/-}$/VDR$^{\Delta/\Delta}$ mutants are characterized by hypercalciuria (Andrukhova *et al*, 2014), and, as shown in the current study, are not consistently hypokalemic and have normal urinary pH. It is likely that the reason for these discrepancies is that FGF23 signaling regulates WNK4 activity. WNK4 is involved in the membrane transport and activation of not only NCC, but also of other ion channels such as TRPV5 and ROMK1 in the distal nephron (Ring *et al*, 2007; Andrukhova *et al*, 2014). Therefore, *Fgf23*$^{-/-}$/VDR$^{\Delta/\Delta}$ and *Kl*$^{-/-}$/VDR$^{\Delta/\Delta}$ compound mutants develop a more complex phenotype than Gitelman's syndrome.

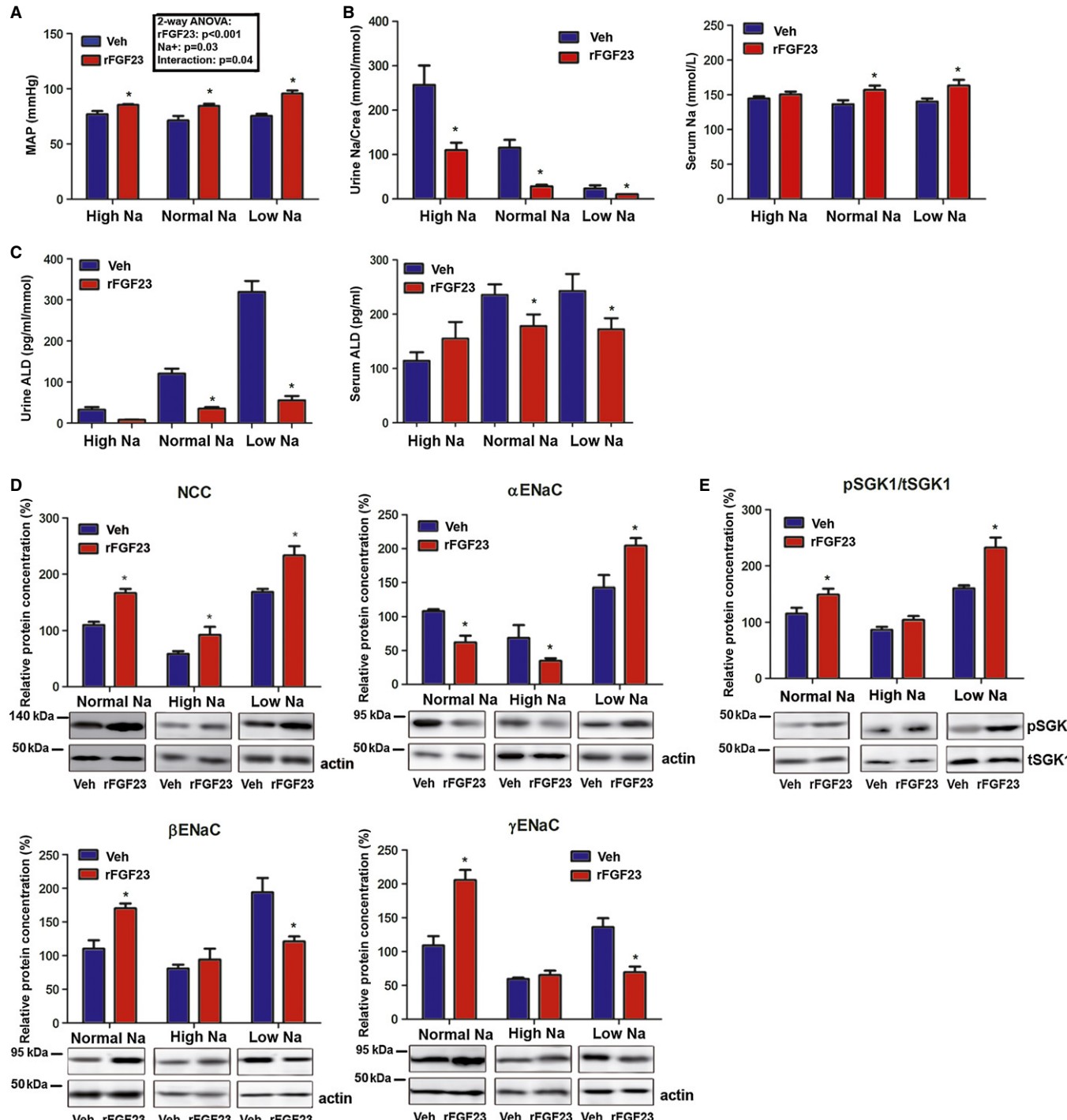

**Figure 7.  Dietary Na+ modulates the effects of rFGF23 on blood pressure and renal Na+ handling.**

A    Mean arterial blood pressure (MAP) of 3-month-old male wild-type mice treated for 5 days with vehicle (Veh) or 10 µg rFGF23 per mouse per day on high (High Na), normal (Normal Na), and low (Low Na) sodium diets ($n = 6–7$, Students $t$-test, * $P < 0.05$ versus vehicle). Inset shows results of two-way ANOVA.

B, C    Urinary Na+ excretion corrected by urinary creatinine (Crea), serum Na+ concentration ($n = 6–7$, Students $t$-test * $P < 0.05$ versus vehicle) and urinary aldosterone corrected by urinary creatinine (Crea) and serum aldosterone concentrations ($n = 6–7$, Students $t$-test, * urine $P < 0.0005$, serum $P < 0.05$ versus vehicle) after 5 days of treatment of 3-month-old male wild-type mice with vehicle or rFGF23 (10 µg per mouse per day) on high, normal and low sodium diets.

D, E    Western blotting quantification of NCC, α-ENaC, β-ENaC and γ-ENaC protein expression in renal cortical total membrane fractions ($n = 4–5$, Students $t$-test, *$P < 0.05$ versus vehicle), and ratio of phospho-SGK1 versus total-SGK1 protein expression in kidney total homogenates of 3-month-old wild-type mice on high, normal, and low sodium diets treated for 5 days with vehicle or rFGF23 ($n = 4–5$, Students $t$-test, *$P < 0.01$ versus vehicle). Data represent mean ± s.e.m.

Source data are available for this figure.

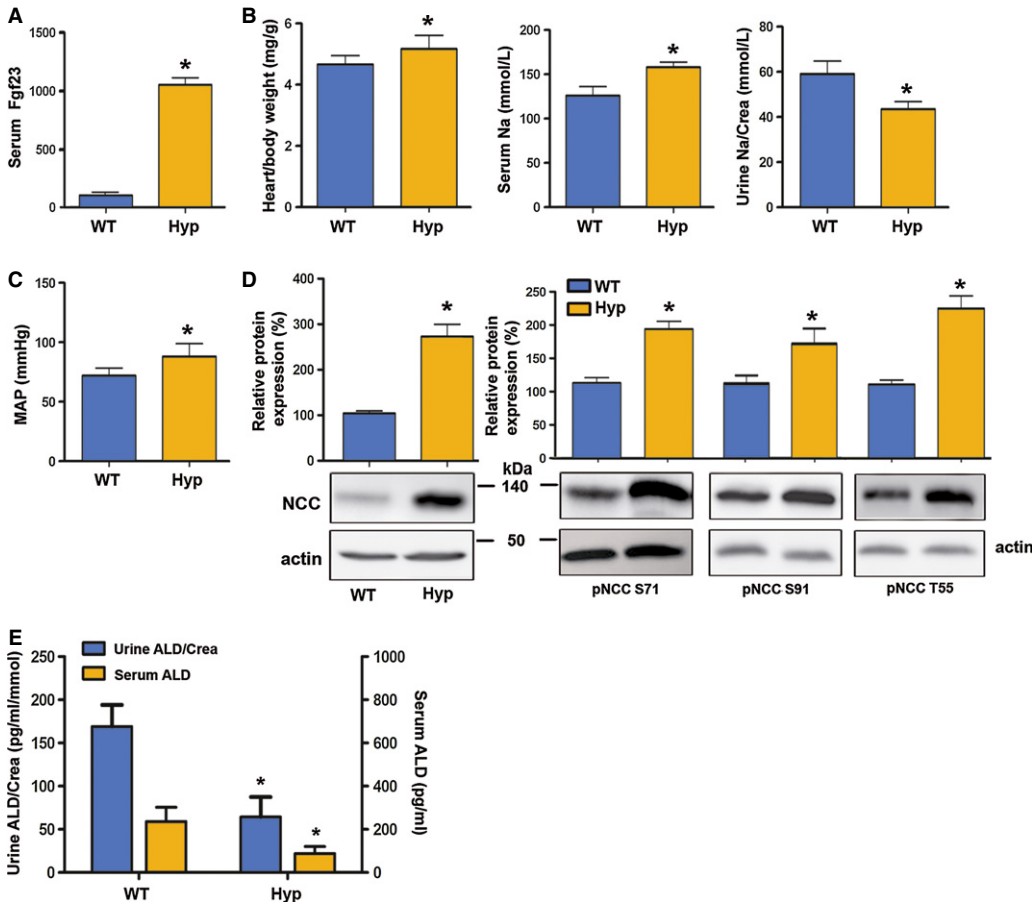

**Figure 8.** *Hyp* mice show hypertension and increased NCC expression and phosphorylation.

A–E   Serum intact Fgf23 concentration (A, *n* = 8–9, Students *t*-test, *P = 0.0001 versus vehicle); heart-to-body weight ratio, serum Na+ and urinary Na+ excretion corrected by urinary creatinine (Crea) (B); mean arterial blood pressure (C, *n* = 8–9, Students *t*-test, *P < 0.05 versus vehicle); Western blotting quantification of renal NCC membrane expression and NCC phosphorylation at Ser71, Ser91, and Thr55 (pNCC S71, pNCC S91, pNCC T55) (*n* = 8–9) (D); and urinary aldosterone corrected by urinary creatinine and serum aldosterone concentrations in 3-month-old male wild-type (WT) and *Hyp* mice (*n* = 8–9, Students *t*-test, *P < 0.05 versus vehicle) (E). Data represent mean ± s.e.m.

Source data are available for this figure.

The physiological function of WNK4 in the regulation of distal renal tubular NCC membrane abundance is still controversial. It was previously thought that the WNK4 mutations found in patients with pseudohyperaldosteronism type II (PHAII, an autosomal dominant disease characterized by hypertension, hyperkalemia, and metabolic acidosis) are loss-of-function mutations and that WNK4 activation inhibits the membrane transport of NCC (Yang *et al*, 2003). However, more recent studies in mice with targeted disruption of the *Wnk4* gene suggest that WNK4 is actually a positive regulator of NCC membrane abundance and function (Ohta *et al*, 2009; Castaneda-Bueno *et al*, 2012). This notion is also supported by our data which suggest that FGF23-induced serine phosphorylation of WNK4 *increases* the complex formation between NCC and WNK4, and the distal tubular membrane abundance of NCC.

The current study has shown that α- versus β- and γ-ENaC subunits are reciprocally regulated in loss- and gain-of-Fgf23 function models. In addition, our data suggest that this regulation is an indirect, aldosterone-mediated process. Aldosterone regulates the

abundance of the ENaC complex by selectively upregulating the α-subunit (May *et al*, 1997; Masilamani *et al*, 1999). In agreement with this notion, urinary aldosterone and renal α-ENaC expression were higher in *Fgf23*−/−/VDR^Δ/Δ and *Kl*−/−/VDR^Δ/Δ compound mutants versus VDR single mutants, whereas serum aldosterone and renal α-ENaC expression were lower in rFGF23-treated and *Hyp* mice versus vehicle-treated and wild-type mice, respectively. Conversely, the full-length β- and γ-subunits were downregulated in loss-of-Fgf23 function models and upregulated in gain-of-Fgf23 function models. The C-terminal proline-rich motifs of the β- and γ-subunits of the ENaC complex interact with the ubiquitin ligase Nedd-4 and are involved in ubiquitination and degradation of ENaC (Lee *et al*, 2009). Aldosterone has been shown to induce proteolytic cleavage of the γ-subunit (Masilamani *et al*, 1999). Therefore, the reciprocal regulation of α- versus full-length β- and γ-ENaC subunits observed in our loss- and gain-of-function models can likely be explained by the concomitant changes in aldosterone signaling.

A surprising finding in our study was that a low Na+ diet augmented the rFGF23-induced increase in arterial blood pressure

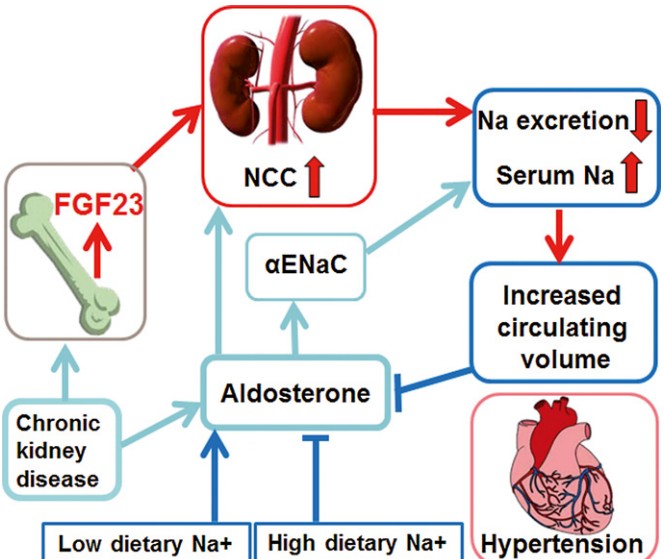

**Figure 9.  Proposed model of FGF23-mediated bone-kidney-heart axis.**
Increased circulating FGF23 augments distal renal tubular NCC expression and activity which leads to renal $Na^+$ retention, volume expansion, hypertension, and heart hypertrophy. As a counter-regulatory mechanism, hypernatremia and increased blood volume decrease aldosterone secretion from adrenal glands, leading to a downregulation of renal α-ENaC expression. A low sodium diet augments the hypertensive effect of increased FGF23 signaling in this model, because it interferes with the counter-regulatory downregulation of aldosterone. Similarly, in chronic kidney failure FGF23 and aldosterone signaling pathways are concurrently activated, potentially leading to a stimulation of both NCC and α-ENaC-driven $Na^+$ reabsorption mechanisms in renal distal tubules, and subsequent augmentation of the FGF23-induced volume expansion, hypertension, and heart hypertrophy.

as compared to rFGF23-treated mice on a normal or high $Na^+$ diet. Interestingly, urinary aldosterone excretion and renal NCC and α-ENaC expression were inversely correlated with dietary $Na^+$ in rFGF23-treated mice. Therefore, we hypothesized that higher aldosterone levels on the low $Na^+$ diet might further enhance the rFGF23-induced SGK1 activation, leading to higher distal renal tubular expression of NCC and α-ENaC and consequently higher renal $Na^+$ reabsorption. Indeed, we found that SGK1 phosphorylation was inversely associated with dietary $Na^+$ in rFGF23-treated mice. Our finding that aldosterone and FGF23 signaling converge on SGK1 and interact in the regulation of NCC- and ENaC-driven $Na^+$ reabsorption in the distal nephron may have important implications for clinical medicine. SGK1 is a central molecule in the regulation of renal $Na^+$ handling and in the pathophysiology of hypertension and renal fibrosis (Lang et al, 2009). If extrapolated to humans, our data would predict that in situations where circulating concentrations of both aldosterone and intact FGF23 are elevated, such as in chronic kidney disease, aldosterone may amplify the effects of FGF23 on $Na^+$ retention. Activation of the renin-angiotensin-aldosterone system (RAAS) is a typical finding in chronic kidney disease (Lattanzio & Weir, 2010). Conversely, based on our data, RAAS inhibition as a therapeutic intervention may also modulate the $Na^+$-conserving function of FGF23 in the kidney. Similar to FGF23, aldosterone can activate NCC through a signaling mechanism involving SGK1, WNK4, and STE20/SPS-1-related proline/alanine-rich kinase (SPAK) (Rozansky et al, 2009; van der Lubbe et al, 2012; Ko et al,

2013). Therefore, aldosterone and FGF23 have synergistic effects on NCC activation. Although ERK1/2 activation has also been implicated in NCC ubiquitination and degradation in some cellular models (Ko et al, 2007, 2010), our study has clearly established that FGF23 increases the membrane abundance and activates NCC through a ERK1/2-SGK1-WNK4 signaling pathway. NCC is mainly expressed in the entire distal convoluted tubule, whereas ENaC is expressed in the late distal convoluted tubule (DCT2), the connecting tubule, and the collecting duct (Nesterov et al, 2012). Whether the crosstalk between aldosterone and FGF23 signaling involves only DCT2, where NCC and ENaC are co-expressed, or also other nephron segments, is currently unclear. In addition, it is currently unclear why SGK1 activation by FGF23 signaling does not directly upregulate α-ENaC expression in distal tubules. It is conceivable in this context that different SGK1 activators such as FGF23 or aldosterone result in different phosphorylation patterns of downstream molecules such as Nedd4 (Flores et al, 2003) due to specific modulation of the activity of additional protein kinases or phosphatases.

Hyp mice are a model of XLH in humans. Our data showed that chronically elevated circulating levels of intact Fgf23 levels in Hyp mice lead to $Na^+$ retention, hypertension, and heart hypertrophy through increased expression of NCC. To the best of our knowledge, data about $Na^+$ homeostasis are not available in XLH patients. Notably, XLH patients show a high incidence of left ventricular hypertrophy, although they are not hypertensive (Nehgme et al, 1997). In analogy to Hyp mice, it is conceivable that chronically increased plasma volume due to FGF23-induced $Na^+$ retention may contribute to the development of left ventricular hypertrophy in XLH patients.

FGF23 is a protective hormone against the untoward biological consequences of hyperphosphatemia. Hyperphosphatemia is a major risk factor for vascular calcification in patients with chronic renal disease (Scialla et al, 2013) and cardiovascular disease in normal subjects (Dhingra et al, 2007). Therefore, in a hyperphosphatemic situation, it may make biological sense to couple increased phosphaturia with renal $Na^+$ conservation and volume expansion in order to additionally "dilute" extracellular phosphate to prevent vascular calcification. However, the downside of this putative protection mechanism may be that chronic gain of FGF23 function causes volume expansion, hypertension, and heart hypertrophy through upregulation of distal renal tubular NCC. Thus, our findings may provide a mechanistic explanation why circulating FGF23 is associated with cardiovascular risk and mortality in patients with CKD, and may reposition NCC blockers such as thiazides in the therapy of CKD and of other conditions characterized by elevated intact circulating FGF23. Elevated aldosterone levels may additionally augment the effects of FGF23 on $Na^+$ retention in CKD patients. A major task for the future is to determine the detailed molecular mechanisms involved in the crosstalk between aldosterone and FGF23 signaling in distal renal epithelium. Moreover, based on our data, it is conceivable that high dietary phosphate intake might predispose to the development of CKD and hypertension through augmented FGF23-induced SGK1 activation and $Na^+$ retention also in the normal population and that aldosterone might modulate this effect. Interestingly, a recent epidemiologic study in almost 14,000 US adults reported that higher dietary $Na^+$ intake was associated with lower odds of CKD (Sharma et al, 2013). Based on our finding that the hypertensive effects of FGF23 are suppressed by high

dietary $Na^+$ (Fig 9), the interaction between phosphate and $Na^+$ intake may be an important determinant of cardiovascular and kidney health in humans.

## Materials and Methods

### Animals

All animal procedures were approved by the Ethical Committees of the University of Veterinary Medicine Vienna and of the local government authorities. Heterozygous $VDR^{+/\Delta}$ (Erben *et al*, 2002) were mated with heterozygous $Fgf23^{+/-}$ (Sitara *et al*, 2004), and heterozygous $Kl^{+/-}$ (Lexicon Genetics, Mutant Mouse Regional Resource Centers, University of California, Davis, CA, USA) mutant mice to generate double heterozygous animals. $Fgf23^{+/-}/VDR^{+/\Delta}$ and $Kl^{+/-}/VDR^{+/\Delta}$ mutant mice on C57BL/6 background were interbred to generate WT, $VDR^{\Delta/\Delta}$, $Kl^{-/-}$, $Fgf23^{-/-}$ and compound $Fgf23^{-/-}/VDR^{\Delta/\Delta}$ and $Kl^{-/-}/VDR^{\Delta/\Delta}$ mutant mice. Genotyping of the mice was performed by multiplex PCR using genomic DNA extracted from tail as described (Hesse *et al*, 2007; Anour *et al*, 2012). The mice were kept at 24°C with a 12/12-h light/dark cycle and were allowed free access to a rescue diet and tap water. The rescue diet (Ssniff, Soest, Germany) containing 2.0% calcium, 1.25% phosphorus, 20% lactose, and 600 IU vitamin D/kg was fed starting from 16 days of age. This diet has been shown to normalize mineral homeostasis in VDR-ablated mice (Li *et al*, 1998; Erben *et al*, 2002). *Hyp* mice were kept on a normal mouse diet and genotyped by PCR analysis. For some experiments, male C57BL/6 mice on a normal mouse chow (Ssniff, Soest, Germany) were used. In 3- and 9-month-old mice, urine was collected in metabolic cages during a 12-h period overnight from 7 p.m. to 7 a.m. In 4-week-old mice, spontaneous urine was collected before necropsy. Some 3-month-old mice received daily intraperitoneal injections of vehicle (phosphate-buffered saline with 2% DMSO), 10 μg recombinant human FGF23 R176/179Q (rFGF23, kindly provided by Amgen, Thousand Oaks, CA, USA) per mouse, or 25 mg/kg chlorothiazide (CTZ, Sigma) for 5 days, and were killed 8–12 h after the last injection. For the experiment with different $Na^+$ diets, 3-month-old male C57BL/6 mice were allowed to adapt to control (0.2%), low (0.05%), and high (4%) $Na^+$ diets (Ssniff, Soest, Germany) for a 3-day period. Starting at day 4, mice on the different $Na^+$ diets were treated with rFGF23 (10 μg/day/mouse) or vehicle for 5 days. At necropsy, the mice were exsanguinated from the abdominal V. cava under anesthesia with ketamine/xylazine (67/7 mg/kg i.p.) for serum collection. In some mice, food consumption was calculated as the average intake of rescue diet over 7 days. Food intake and body weight were measured every 24 h.

### Histology

Hearts were fixed in 40% ethanol for 48 h, embedded in paraffin, and routinely stained with hematoxylin/eosin.

### Serum and urine biochemistry

Serum and urinary sodium, potassium, phosphorus, and creatinine were analyzed on a Hitachi 912 Autoanalyzer (Boehringer Mannheim) or on a Cobas c111 analyzer (Roche). Serum and urinary aldosterone were determined by ELISA (NovaTec). Plasma renin activity was measured by RIA (GammaCOAT, DiaSorin). Serum intact Fgf23 was assessed by ELISA (Kainos).

### Immunohistochemistry

For immunohistochemistry, 5-μm-thick paraffin sections of paraformaldehyde (PFA)-fixed kidneys were prepared. Before immunofluorescence staining, dewaxed sections were pretreated with blocking solution containing 5% normal goat serum in PBS with 0.1% bovine serum albumin and 0.3% Triton X-100 for 60 min. Without rinsing, sections were incubated with polyclonal rabbit anti-NCC (Millipore, 1:500) or anti-α-ENaC (Novus Biologicals, 1:500) antibodies at 4°C overnight. After washing, sections were incubated for 1.5 h with goat anti-rabbit Alexa 488 or goat anti-rabbit Alexa 568 secondary antibodies (Invitrogen, 1:400), respectively. Controls were performed by omitting primary antibodies. The slides were analyzed on a Zeiss LSM 510 Axioplan 2 confocal microscope equipped with a 63 × oil immersion lens (NA 1.3). Individual fluorochromes were simultaneously excited by lasers at 488- and 543-nm wavelengths with appropriate filter sets for the emitted light to avoid crosstalk. Images were merged using Adobe Photoshop.

### Total cell membrane isolation

Mouse kidney cortex was homogenized in a homogenizing buffer [20 mM Tris (pH 7.4/HCl), 5 mM $MgCl_2$, 5 mM $NaH_2PO_4$, 1 mM ethylenediaminetetraacetic acid (pH 8.0/NaOH), 80 mM sucrose, 1 mM phenyl-methylsulfonyl fluoride, 10 μg/ml leupeptin, and 10 μg/ml pepstatin] and subsequently centrifuged for 15 min at 4,000 *g*. Supernatants were transferred to a new tube and centrifuged for an additional 30 min at 16,000 *g*.

### Isolation of distal tubular segments

Renal distal tubules were isolated as reported previously (Andrukhova *et al*, 2012, 2014). In brief, murine kidneys were perfused with sterile culture medium (Ham's F12; GIBCO) containing 1 mg/ml collagenase (type II; Sigma) and 1 mg/ml pronase E (type XXV, Sigma) at pH 7.4 and 37°C. The cortical tissue was dissected in small pieces and placed at 37°C in sterile Ham's F12 medium containing 0.5 mg/ml collagenase II and 0.5 mg/ml pronase E for 15 min with vigorous shaking. After centrifugation at 3,000 rpm for 4 min, the enzyme-containing solution was removed, and tubules were resuspended in ice-cold medium. Individual distal tubule segments were identified based on morphology in a dissection microscope at ×25–40 magnification by their appearance and dimensions. To rule out contamination with proximal tubules, we performed purity and quality controls, using mRNA expression of distal (TRPV5, calbindin 28k) and proximal (NaPi-2a, NaPi-2c) tubule-specific genes (Andrukhova *et al*, 2012). Distal tubular segments from wild-type, $Kl^{-/-}$ and $Fgf23^{-/-}$ mice were incubated with vehicle (PBS) or rFGF23 (100 ng/ml) and/or 10 ng/ml of the specific SGK1 kinase inhibitor GSK 650394 (Axon Medchem), or 10 ng/ml of the ERK1/2 inhibitor PD184352 (Sigma) for 2 h. Protein samples were collected for Western blotting analysis in lysis buffer.

## Western blot

Kidney cortex homogenates or total cell membrane preparations were solubilized in Laemmli sample buffer, fractionated on SDS–PAGE (30 μg/well) and transferred to a nitrocellulose membrane (Thermo Scientific). Immunoblots were incubated overnight at 4°C with primary antibodies including polyclonal rabbit anti-NCC (1:3,000, Millipore), rabbit anti-phospho-NCC Ser71 (pNCC S71; 1:1,000), anti-phospho-NCC Ser91 (pNCC S91; 1:1,000), anti-phospho-NCC Thr 55 (pNCC T55; 1:1,000) (generous gifts of Dario R. Alessi, University of Dundee, Dundee, UK), anti-α-ENaC (Novus Biologicals, 1:1,000), anti-β-ENaC and anti-γ-ENaC (Antikoerperonline.com, 1:1,500), anti-WNK4 (1:2,000, Novus Biologicals), and monoclonal mouse anti-β-actin (1:5,000, Sigma) in 2% (w/v) bovine serum albumin (BSA, Sigma) in a TBS-T buffer [150 mM NaCl, 10 mM Tris (pH 7.4/HCl), 0.2% (v/v) Tween-20]. After washing, membranes were incubated with horseradish peroxidase-conjugated secondary antibodies (Amersham Life Sciences). Specific signal was visualized by ECL kit (Amersham Life Sciences). The protein bands were quantified by Image Quant 5.0 software (Molecular Dynamics). The expression levels were normalized to Ponceau S stain.

## Co-immunoprecipitation

Kidney cortex homogenate protein samples (1 mg) were incubated with 2 μg of anti-WNK4 (Novus Biologicals), anti-phosphoserine (Alpha Diagnostics), or anti-NCC (Millipore) antibody at 4°C overnight. The immune complexes were captured by adding 50 μl Protein A or G agarose/sepharose beads (Santa Cruz Biotechnology) and overnight incubation at 4°C with gentle rocking. The immunoprecipitates were collected by centrifugation at $1,000 \times g$ for 5 min at 4°C and washed for four times in PBS, each time repeating the centrifugation step. After the final wash, the pellets were suspended in 40 μl of electrophoresis sample buffer and boiled for 2–3 min. Western blot analysis was performed as described above using a primary anti-NCC, anti-phosphoserine, or anti-WNK4 antibody.

## Central arterial and central venous pressures measurements

Central arterial pressure and central venous pressure (CVP) were assessed using a SPR-671NR pressure catheter (1.4F, Millar Instruments, Houston, TX, USA). Central arterial pressure measurements were performed under 1.5% isoflurane anesthesia by inserting the catheter into the ascending aorta via the carotid artery. In addition to the central arterial pressure analysis, central venous pressure was measured by inserting the catheter into the internal jugular vein in experiments with 5 days of treatment with rFGF23 or vehicle. Pressure was recorded over 5 min and traces were analyzed using LabchartPro software and a blood pressure module. CVP was calculated as the average between pressure values of the ascending "a" wave and descending "x" wave determined from at least 5 cardiac cycles.

## Blood volume measurements

The blood volume was determined from plasma volume and hematocrit as described (Lee & Blaufox, 1985). Plasma volume was

determined by Evans blue dye dilution (Barron *et al*, 1984). Briefly, 20 μl of a 0.4% (wt/vol) solution of Evans blue (Sigma) in sterile physiological saline was injected into a tail vein. Blood samples (10 μl) were collected at 10 min and 30 min after injection to measure disappearance kinetics. Tubes were centrifuged and the hematocrit was recorded. Evans blue concentration in the plasma was measured in duplicate as optical density using the 2-wavelength method.

## RNA isolation and quantitative RT-PCR

Shock-frozen hearts were homogenized using TRI Reagent (Molecular Research Center). Total RNA was extracted with phenol/chloroform, precipitated using isopropanol, and then treated with RQ1 RNase-free DNase (Promega). RNA purity and quality was determined spectrophotometrically (BioPhotometer; Eppendorf). After first-strand cDNA synthesis (iScript cDNA Synthesis Kit, Bio-Rad), quantitative RT-PCR was performed on a Rotor-Gene™ 6000 (Corbett Life Science) using SsoFast™ EvaGreen PCR kit (Bio-Rad). A melting curve analysis was done for all assays to make sure that only a single PCR product was amplified. Primer sequences are available on request. Efficiencies were examined by standard curve. Gene expression data were corrected for efficiency and normalized to ornithine decarboxylase antizyme-1 (*Oaz1*) as house-keeping gene.

## Intracellular Na⁺ imaging

Longitudinal 300-μm-thick live slices of freshly isolated kidneys were prepared using a Leica VT1000 Vibratome (Leica Microsystems). For the preparation of 10 mM stock solution of the sodium-sensitive dye SBFI (Molecular Probes), SBFI was diluted in DMSO (Merck Millipore International) and 20% Pluronic (Merck Millipore International). The kidney slices were incubated for 60 min at 37°C with 2 μM SBFI (stock solution diluted 1:5,000 with cell culture medium). Thereafter, the slices were washed two times for 20 min each in 0.1M PBS. Some kidney slices were incubated *in vitro* with rFGF23 (100 ng/ml) or vehicle (PBS). For visualization of intracellular Na⁺ content, SBFI was excited by a Ti:sapphire laser (Chameleon, Coherent Inc.) at 820 nm. Images (512 × 512 pixels) were acquired every 30 sec at a depth of 60–80 μm. For the inhibition of NCC activity, tissue slices were incubated with 10 μM of chlorothiazide (CTZ, Sigma) or vehicle (PBS + 1% ethanol) at 37°C, 5% $CO_2$/ 95% air humidified atmosphere for 30 min. Fluorescence images were analyzed using Image J software. The whole epithelial layer of the distal tubules was selected by manually drawing the region of interest (ROI) to quantify intracellular Na⁺ levels. Fluorescence intensity was quantified in 4–9 ROIs per image, and the ratio between the fluorescence intensity and the ROI area was calculated for each tubule. This ratio was used for all subsequent calculations.

## Statistical analyses

Statistics were computed using SPSS for Windows 17.0. The data were analyzed by two-sided t-test (2 groups) or one-way analysis of variance (ANOVA) followed by Student-Newman-Keuls (SNK) multiple comparison test (> 2 groups). In addition, arterial blood pressure data from the Na⁺ diet experiment were analyzed by two-way ANOVA, assessing the influence of the diet and of rFGF23

**The paper explained**

**Problem**

Fibroblast growth factor-23 (FGF23) is a hormone secreted by bone cells in response to increased extracellular phosphate and vitamin D. FGF23 in turn stimulates renal phosphate excretion and suppresses vitamin D hormone synthesis as part of a negative feedback loop between bone and kidney. In patients with chronic kidney disease (CKD), the declining kidney function leads to decreased renal phosphate excretion, increased blood phosphate levels, and subsequently elevated FGF23 serum levels. Interestingly, prospective and cross-sectional clinical studies have shown that circulating FGF23 is positively and dose dependently associated with CKD progression, cardiovascular risk factors such as left ventricular hypertrophy, vascular calcifications, and mortality in CKD patients. The molecular mechanism underlying these associations has so far remained elusive.

**Results**

Here, we show that FGF23 is a direct regulator of the sodium-chloride channel NCC in distal renal tubules. This channel has a crucial role in the reabsorption of sodium from renal tubules. Mice lacking Fgf23 or its co-receptor Klotho showed lower expression of NCC, leading to renal sodium wasting, reduced plasma volume, and lower blood pressure. Conversely, injection of recombinant FGF23 into normal mice resulted in upregulation of renal NCC expression, renal sodium retention, plasma expansion, hypertension, and heart hypertrophy. Co-treatment with the NCC channel blocker chlorothiazide abrogated the FGF23-induced volume expansion and increase in blood pressure. Intriguingly, a low sodium diet aggravated the hypertensive effects of recombinant FGF23 in normal mice, probably because intracellular signaling of FGF23 and of the other major sodium-conserving hormone aldosterone converge on the same molecules in distal renal tubules.

**Impact**

Our study identifies FGF23 as a sodium-conserving hormone. Because sodium homeostasis is tightly coupled to volume regulation and blood pressure, our paper may explain why FGF23 is associated with cardiovascular risk and mortality in CKD patients. In addition, our study may reposition NCC blockers such as thiazide diuretics in the therapy of CKD and of other conditions characterized by elevated circulating FGF23. The novel link between phosphate and sodium homeostasis may also have important implications for the general population. Based on our findings, a high dietary phosphate intake might predispose to the development of hypertension.

treatment as well as their two-way interaction. *P* values of less than 0.05 were considered significant. The data are presented as the mean ± s.e.m.

**Supplementary information** for this article is available online: http://embomolmed.embopress.org

## Acknowledgments

We thank C. Bergow for excellent technical assistance and William G. Richards for critical reading of the manuscript. Recombinant FGF23 was a generous gift of Vicky Shalhoub, Amgen Inc. Thousand Oaks, CA. The anti-phospho-NCC antibodies were generous gifts of Dario R. Alessi, University of Dundee, Dundee, UK. We are grateful to Manoocher Soleimani, Cincinnati University, OH, USA, and to Gerardo Gamba, Molecular Physiology Unit, Mexico City, Mexico, for providing kidneys from NCC- and WNK4-knockout mice, respectively. This work was supported by a grant from the Austrian Science Fund (FWF P24186-B21) to R.G.E. O.A. was supported by a postdoctoral fellowship of the University of Veterinary Medicine Vienna.

## Author contributions

OA, AS, EEP, and RGE conceived and designed the experiments; OA, SS, AS, and UZ performed experiments and analyzed the data; OA and RGE wrote the manuscript; VS and BL provided important tools; OA, SS, AS, VS, BL, EEP, and RGE discussed and reviewed the manuscript.

## Conflict of interest

VS was an employee of Amgen, Inc. The other authors declare no conflict of interest.

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
