## [Review Process File · EMBO Molecular Medicine]

FGF23 Regulates Renal Sodium Handling and Blood Pressure

Olena Andrukhova, Svetlana Slavic, Alina Smorodchenko, Ute Zeitz, Victoria Shalhoub, Be-ate Lanske, E.E. Pohl, and Reinhold G. Erben

Corresponding author: Reinhold Erben, University of Veterinary Medicine

Review timeline:

Submission date:	31 December 2012
Editorial Decision:	30 January 2013
Additional author correspondence :	21 February 2013
Additional editorial correspondence :	21 February 2013
Resubmission:	25 November 2013
Editorial Decision:	22 December 2013
Revision received:	06 February 2014
Editorial Decision:	14 February 2014
Revision received:	27 February 2014
Accepted:	05 March 2014

Transaction Report:

Editor: Roberto Buccione

1st Editorial Decision

30 January 2013

Thank you for the submission of your manuscript to EMBO Molecular Medicine. We have now heard back from the three Reviewers whom we asked to evaluate your manuscript.

As you will see, the Reviewers, while acknowledging the potential interest of your work, all point to several issues of a fundamental nature that, I am afraid, preclude publication of the manuscript in EMBO Molecular Medicine. I feel it unnecessary to report each point in detail here as they are clearly stated by the Reviewers, but I will mention a few salient items.

Reviewer 1 for instance has technical concerns related to the interpretation of the results obtained with metabolic cages and the duration of urine sampling. In addition, s/he finds it difficult to envisage that FGF23 stimulation of NCC can provoke hypertension in chronic kidney disease (CKD) patients without further experimentation. Reviewer 1 also challenges, based on the data, the interpretation that genetic ablation of FGF23 increases natriuresis and suggests that sodium intake between the genotypes should be proven to be identical to support this claim. Reviewer 2 also notes that urine aldosterone concentration is an incorrect parameter to consider given the experimental conditions (see also the technical concerns mentioned above). Reviewer 1 also lists other major critical points.

Reviewer 2 shares many of the concerns expressed by Reviewer 1, including the interpretation and experimental support for your conclusions on the double knockout animal model. S/he also

challenges the effect of recombinant FGF23 on NCC as proposed based on insufficient experimental evidence and maintains that, as a consequence, the experiments with chlorothiazide do not clarify the issue. I should mention that Reviewer 1 has the same specific concern. Reviewer 2 is also concerned about the appearance of NCC on the western blots; again Reviewer 1 shares exactly the same preoccupation.

Reviewer 3, while more positive, does point out some important issues that add up to the others. For instance, s/he does not feel that the conclusion that NCC mediates the cardiovascular effects of increased FGF23 up-regulation is sufficiently supported by the data. Also, Reviewer 3 agrees with Reviewers 1 and 2 that the data on urinary sodium excretion should take into account urine evaporation. Reviewer 3 also lists other critical points, which in our opinion are of essential importance.

I am sure you will understand that all considered, the issues are too many and too fundamental and thus leave us no choice but to return the manuscript to you at this stage.

I am sorry to disappoint you on this occasion. I hope, however, that the referee comments will be helpful in your continued work in this area.

***** Reviewer's comments *****

Reviewer #1 (Remarks):

In this study Andrukhova et al. did use different mouse models in order to test the effects of fgf23 on the distal tubule, Na transport handling and blood pressure. They propose that fgf23 stimulates NaCl absorption via the NaCl co transporter NCC and thereby increases vascular volume and raises blood pressure. The question is interesting since high levels of circulating FGF23 has been associated with the onsets of cardiovascular events in chronic kidney disease patients. Although the reviewer found some merits to this study he also feels that some serious concerns can be raised about the experimental design and data analyses. As a consequence, I am not sure that the data presented here fully support the claim of the authors

Major:

1) It is difficult to conceive that stimulation of NCC by fgf23 can provoke hypertension in CKD patients since the patients with elevated fgf23 have generally quite severe renal failure and thus have a major reduction in functional renal mass. The clinical relevance of the present study would be much more convincing if the authors could show that patients with normal renal function and elevated fgf23 are prone to hypertension or salt retention. These patients exist (oncogenic osteomalacia for example). The reviewer is not aware that these patients are characterized by hypertension. Is it the case? Do you have access to these patients?

2) The demonstration that fgf23 promotes renal salt retention or that ablation of fgf23 causes renal salt wasting has been performed using mouse models in which fgf23 was absent (fgf23^{-/-}) or injected. The observation that genetic ablation of fgf23 increases natriuresis in 9 month-old fgf23^{-/-}/VDR^{-/-} is rather surprising. In fact, while renal salt wasting can be unmasked in acute maneuvers (acute salt restriction) in chronic situations, sodium has to be balanced or mice would die. Therefore, I do not agree with the interpretation of the authors. An increase in steady state natriuresis doesn't reflect a salt losing nephropathy but is rather the reflect of an increased sodium input into the body. Authors should prove that sodium intake was identical between genotypes.

3) Plasma and urine aldosterone are important parameters reflecting extracellular volume variations. However, aldosterone is also modulated by potassium homeostasis. It would be worth to provide a second marker such as plasma renin activity or atrial natriuretic factor. Authors state that urine aldosterone measurement is more accurate because urine is collected over a long period, and that this buffers differences that can be observed on plasma sampling which can be altered by stress or sampling itself. I fully agree with this statement. However, the authors provide the reader not with aldosterone output over a 24h period but with urine aldosterone concentration. This is obviously not

correct. To reflect aldosterone secretion the authors have to show 24h output or aldo/creatinine ration. This is particularly critical since urine output is likely to be different between genotype.

4) More generally, the authors should be more cautious when interpreting results from metabolic cages if urine output differs. Indeed, sampling of urine in metabolic cages is particularly difficult in mouse experiments because the total amount of urine voided over 24h is very small (when compared to rats for instance) and a large amount of urine evaporates in the collecting system. Obviously the error consecutive to this evaporation is minimal when urine output is more important. Thus, the authors should consider that some differences in sodium output might be explained by urine collection differences due to differences in urinary volume. Normalizing urine parameter to urine creatinine is should address this potential problem.

5) 12h hour urine sampling is also not really correct for this kind of study. Urinary sodium excretion varies along day and night. Circadian rhythm has a major effect on urinary sodium secretion. As an example, circadian rhythm might be affected in some of the knock out animals and explain the difference in natriuresis. To limit those effects, 24h urinary measurements should be performed.

6) The reviewer is puzzled to some extent by the look of NCC band in the western blot experiments. Glycosylated NCC usually migrates as multiple bands (or smear) around 130kD and above. It doesn't seem to be the same for the NCC western blot (WB) shown here. Therefore, authors should provide full blots to reviewers and indicate molecular weight of the WB bands. In general it is not correct to show as shown here only one single or few representative blots, and it is also not correct to cut the images just around the band of interest.

7) aENaC is part of a multimeric complex forming the channel. Measurements of Beta and gamma subunit renal abundance are also necessary for interpretation. Moreover, a and g EnaC are processed as different polypeptides that should be quantified separately.

8) The effect of Fgf23 on heart remodeling is already known as acknowledge by the authors. The hypertensive effect of rFgf23 is interesting and is well demonstrated. The authors propose that Fgf23 directly upregulates NCC thereby increasing salt retention. However, western blots test only protein abundance and do not reflect always changes in activity. The effects of fgf23 on NCC activity should be provided. This could be tested in a cell culture model. Alternatively, the changes in natriuresis induced by an acute hydrochlorothiazide (HCTZ) injection might be used to test a potential difference in NCC activity in vivo. Long-term infusion of HCTZ infusion to test NCC stimulation by fgf23 is rather indirect and doesn't reflect necessarily an activation of NCC upon fgf23 treatment. Indeed, many compensatory mechanisms can be triggered during chronic HCTZ treatment and can affect the results of the experiments presented here.

Minor:

It is not very clear in the legends how many animals per group have been used for biological parameters, immunofluorescence and western blot and should be provided for each type of experiments.

Reviewer #2 (Remarks):

Andrukhova et al presents an interesting study suggesting that FGF23 modulates the expression of the renal NaCl cotransporter, NCC of the distal convoluted tubule and consequently the epithelial sodium channel, ENaC of the collecting duct. They show that renal salt reabsorption is reduced in the double knockout for FGF23 and Klotho and conversely that FGF23 administration in wild type mouse causes salt retention.

The major strength of the study is the multiple animal models in which the authors test their hypothesis and that seems to correlate the observations in opposite models. The study, however, has some important weakness that precludes the acceptance of the conclusions.

The study of the double knockout animals FGF23^{-/-} and VDR^{+/+} or KI^{-/-} and VDR^{+/+} is uncompleted as to understand the model. They show that 9 months animals exhibit a remarkable increase in salt excretion, but no other data on salt balance is presented. Are these mice dehydrated? Is blood pressure ok? If not, where do they take the excreted salt from? Are they thus eating more food? Could this be the consequence of the diet administered to these mice? If they were not eating more food, I would expect these mice to be dehydrated and hypotensive.

If the double mutant animals discussed above has a real decrease on NCC one would expect a full Gitelman's syndrome to be developed, so it will be desirable that a complete characterization of these mice is presented. Are they hypokalemic? Are they hypocalciuric? Do they have metabolic alkalosis? Supplemental figure 1 shows that fgf23 double mutant has lower serum K⁺ levels, but not because it is low (is above 4.0) but because it is higher in the controls. In fact, in the double mutant Klotho potassium level is increased.

The antibodies used for NCC and WNK4 are commercially developed and poorly characterized. Most studies using good antibodies for NCC shows a smear band corresponding to glycosylated form of the protein. In this study only a fine band is presenting. In addition, this reviewer believes that in this type of studies it is more convincing when the data from all animals is presented. If figure 1C or 1D was done using between 8 and 12 animals, why not showing all of them to the reader can judge for itself the results. Also, since authors are using a commercial antibody not well characterized, it will be better if they show the entire blot, so the reader can judge the size of the bands that come out with the antibodies. In the present form we have no clue of what size the presented bands are. This is also valid for ENaC, which usually is presented as a double band, so we do not know which band is the one that is presented.

The antibody for WNK4 is also a commercial one that has not been properly studied. In the WNK field it is well known that one problem has been the lack of a truly good anti-WNK4 antibody detecting WNK4 in renal tissue.

The effect of rFGF23 on NCC is not necessary due to direct effect on the cotransporter in DCT. FGF23 could have an effect on other systems, like adrenergic or renin-angiotensin that is known to activate NCC. The experiment with dissected tubules could help in this issue. However, isolated number of DCTs isolated this way is very limited. I suggest doing the following. On the one hand to get DCTs isolated by the COPAS methodology that has been recently mastered by Bindel's group in Netherlands or by Loffing in Zurich, this way the amount of DCTs for study are seriously increased. On the other hand, to repeat the DCT experiment shown in Fig 2F but dissecting the tubules from the FGF23 knockout mice.

Because of this previous comment, the experiments with chlorthalidone does not help because we cannot be sure that rFGF23 is actually activating NCC or activating something else that in turn activates NCC, so thiazide would prevent that. In fact, it is surprising that with only five days of rFGF23 administration is enough to cause even cardiac hypertrophy due to hypertension that is secondary to increased salt reabsorption by NCC.

How the authors conceal their observation that inhibiting ERK1/2 prevents the increased expression of NCC by FGF23 with published evidences that activation of ERK1/2 is associated with decreased expression of NCC (1, 2), so inhibition of ERK1/2 would be expected to be associated with up-regulation of NCC.

1. Ko B, Joshi LM, Cooke LL, Vazquez N, Musch MW, Hebert SC, Gamba G, and Hoover RS. Phorbol ester stimulation of RasGRP1 regulates the sodium-chloride cotransporter by a PKC-independent pathway. *Proc Natl Acad Sci U S A* 104: 20120-20125, 2007.
2. Ko B, Kamsteeg EJ, Cooke LL, Moddes LN, Deen PM, and Hoover RS. RasGRP1 stimulation enhances ubiquitination and endocytosis of the sodium-chloride cotransporter. *Am J Physiol Renal Physiol* 299: F300-F309, 2010.

Reviewer #3 (Remarks):

The present paper describes the undescribed role of FGF23 of regulating Na⁺ homeostasis, systemic blood pressure and heart hypertrophy. By using compound mutant mice lacking Fgf23 or klotho and VDR, the Authors found that double mutant ko mice showed renal Na⁺ wasting with respect to VDR mutants and WT animals, associated with elevated urinary aldosterone, decreased NCC but upregulated ENaC. In contrast, infusion of recombinant FGF23 in control mice induced Na⁺ retention and volume expansion which possibly increased blood pressure and induced heart hypertrophy. Mechanisms underlying the effect of FGF23 on Na⁺ retention and its cardiovascular consequences were explored in vitro and in vivo. The effect was attributed to FGF23-induced upregulation of distal renal tubular NCC.

The paper addresses a novel issue. The experimental design is well done, the methodological approach is sophisticated and the conclusions are in general supported by data. Few additional experiments could help strengthen the conclusions. The following is for the Authors' consideration:

- The Authors showed that cardiovascular effects of increased circulating FGF23 are mediated through the upregulation of distal renal tubular NCC by studies employing the NCC functional blocker chlorothiazide. To finally demonstrate specificity of the action of Na⁺:Cl⁻ cotransporter, NCC ko animals should be used.
- Given the key role of Na⁺ excretion in the described experiments, data on Na⁺ urinary excretion should be expressed as Ur Na/Crea (mmol/mmol) in all the studies (Figure 2A, 2E, 3A) to avoid any bias that reduced Na⁺ excretion could only reflect urine volume reduction after rFGF23 injection.
- Emerging evidence indicates the relationship between Vitamin D, FGF23 and RAAS as nicely summarized by de Borst et al (JAm Soc Nephrol 22:1603, 2011). In conditions of high FGF23 levels, reduced activation of vitamin D upregulates renal renin production. To rule out that the cardiovascular effects of rFGF23 injection could be attributable to RAAS activation, plasma renin activity should be assessed in mice injected with rFGF23.
- The present experiments have been performed with animals fed a normal Na⁺ diet. It should be interesting to study the effect of the injection of rFGF23 in animals fed a low or high Na⁺ diet.
- Serum Na⁺ levels are unchanged in double FGF23 mutant mice but not in double klotho mutant mice (Figure S1). The Authors should explain why.
- WB image of NCC in VDRdelta/delta mice should be replaced (Figure 1D).
- Serum Na⁺ levels are significantly increased in rFGF23 injected animals vs vehicle but the mean values seem comparable.
- Basal values of urinary Na⁺ excretion vary among the experiments (Figure 2A, 2E and supplemental Figure 2, Figure 3A). Same variability is present for heart/body weight values in vehicle treated animals (Figure 2D and Supplemental Figure 3). The Authors should explain such a variability.

Additional author correspondence

21 February 2013

Thank you for sending your decision on this manuscript. Most of the comments are well taken and are really helpful.

In the meantime, we looked at NCC and cardiovascular parameters in Hyp mice which are a model of chronically elevated Fgf23, similar to patients with X-linked hypophosphatemic rickets (XLH) and oncogenic osteomalacia, as suggested by reviewer #1. These new data fully confirm our hypothesis that Fgf23 regulates NCC, and that chronic elevation of endogenous Fgf23 leads to heart hypertrophy. Therefore, we are confident that we have discovered an important new pathophysiological mechanism, which is crucial for an improved understanding of diseases in which FGF23 is chronically elevated.

Since many of the comments of the reviewers are technical in nature, and very easy to answer, would you be willing to consider an improved/revised version of this manuscript (possibly as a new submission), in which we include new data, and address all issues raised?

The related manuscript about Fgf23 regulation of WNK4 and TRPV5 was recently rated as major revision in EMBO J. We are working on the revision.

Additional editorial correspondence

21 February 2013

Thank you for your message.

Rejecting a manuscript, especially after review, is always a joyless affair. Some consolation can ensue when the Authors find the Reviewers' suggestions helpful and well taken. So thank you for mentioning that.

As for your question. You are welcome to submit your revised manuscript, with the understanding that it will be treated as a new submission

I have looked at the Reviewers' comments again. The technical concerns were not minor and thus you are warmly encouraged to take them into full consideration too.

Resubmission

25 November 2013

We would like to thank the reviewers for their constructive critique which has significantly strengthened this manuscript.

Reviewer 1

In this study Andrukhova et al. did use different mouse models in order to test the effects of fgf23 on the distal tubule, Na transport handling and blood pressure. They propose that fgf23 stimulates NaCl absorption via the NaCl co transporter NCC and thereby increases vascular volume and raises blood pressure. The question is interesting since high levels of circulating FGF23 has been associated with the onsets of cardiovascular events in chronic kidney disease patients.

Although the reviewer found some merits to this study he also feels that some serious concerns can be raised about the experimental design and data analyses. As a consequence, I am not sure that the data presented here fully support the claim of the authors

Major:

1. It is difficult to conceive that stimulation of NCC by fgf23 can provoke hypertension in CKD patients since the patients with elevated fgf23 have generally quite severe renal failure and thus have a major reduction in functional renal mass. The clinical relevance of the present study would be much more convincing if the authors could show that patients with normal renal function and elevated fgf23 are prone to hypertension or salt retention. These patients exist (oncogenic osteomalacia for example). The reviewer is not aware that these patients are characterized by hypertension. Is it the case? Do you have access to these patients?

Authors: This is a very good point. We don't have access to TIO patients. However, we included a detailed analysis of Hyp mice as a model of chronically elevated endogenous Fgf23 levels with normal renal function in the revised manuscript (Fig. 8). Hyp mice are a model of X-linked hypophosphatemic rickets (XLH) in humans. We found increased total and phosphorylated NCC protein expression together with decreased urinary sodium excretion in Hyp mice as compared to wild type littermates. In addition, serum sodium concentration, mean arterial pressure (MAP), and heart to body weight ratio were slightly, but significantly increased in Hyp mice, relative to wild-type controls. The data are shown in Fig. 8 of the revised manuscript. We feel that these additional

in vivo data provide strong evidence in support of our hypothesis that FGF23 is associated with extracellular volume expansion, hypertension and heart hypertrophy by regulating NCC expression, independent of the presence of renal disease. We added a para in the discussion on the bottom of page 15 to discuss our findings in *Hyp* mice.

2. The demonstration that fgf23 promotes renal salt retention or that ablation of fgf23 causes renal salt wasting has been performed using mouse models in which fgf23 was absent (fgf23^{-/-}) or injected. The observation that genetic ablation of fgf23 increases natriuresis in 9 month-old fgf23^{-/-}/VDR^{-/-} is rather surprising. In fact, while renal salt wasting can be unmasked in acute maneuvers (acute salt restriction) in chronic situations, sodium has to be balanced or mice would die. Therefore, I do not agree with the interpretation of the authors. An increase in steady state natriuresis doesn't reflect a salt losing nephropathy but is rather the reflect of an increased sodium input into the body. Authors should prove that sodium intake was identical between genotypes.

Authors: We agree with the reviewer that an alternative explanation for the increased renal sodium excretion in Fgf23/VDR and Klotho/VDR mutants could be increased dietary sodium intake. We measured food intake as suggested. However, food intake did not differ between the groups (Suppl. Fig. S1C). All animals were on the same diet containing 0.2% sodium. Therefore, sodium intake did not differ. It is clear that sodium must be balanced in the long run. In order to adapt to reduced NCC expression and the accompanying renal sodium wasting, Fgf23/VDR and Klotho/VDR mutants up regulate aldosterone which in turn results in sodium conservation in a variety of organs. To substantiate our hypothesis, we measured blood pressure and blood volume. As shown in Fig. 2, Fgf23/VDR and Klotho/VDR mutants are hypovolemic and hypotensive. Taken together, sodium is certainly balanced in Fgf23/VDR and Klotho/VDR mutants, however, at the expense of reduced blood volume. We added a paragraph in the results section of the revised manuscript on page 7 to explain this.

3. Plasma and urine aldosterone are important parameters reflecting extracellular volume variations. However, aldosterone is also modulated by potassium homeostasis. It would be worth to provide a second marker such as plasma renin activity or atrial natriuretic factor. Authors state that urine aldosterone measurement is more accurate because urine is collected over a long period, and that this buffers differences that can be observed on plasma sampling which can be altered by stress or sampling itself. I fully agree with this statement. However, the authors provide the reader not with aldosterone output over a 24h period but with urine aldosterone concentration. This is obviously not correct. To reflect aldosterone secretion the authors have to show 24h output or aldo/creatinine ration. This is particularly critical since urine output is likely to be different between genotype.

Authors: As suggested by the reviewer we expressed urinary aldosterone excretion per creatinine in the revised Fig. 1B. In addition, we show urine volume in Suppl. Fig. S1B. The differences in urine output were not very pronounced. Therefore, regardless whether expressed as concentration, per creatinine, or per time, urinary aldosterone excretion was increased in Fgf23^{-/-}/VDR^{ΔΔ} and Klotho^{-/-}/VDR^{ΔΔ} mutants as compared to VDR^{ΔΔ} animals. We also measured plasma renin activity as suggested. However, employing a commercially available assay also used by others (e.g., Bernal-Mizrachi et al. Nat Med. 2003) we found unchanged plasma renin activity in Fgf23^{-/-}/VDR^{ΔΔ} and Klotho^{-/-}/VDR^{ΔΔ} mutants as shown in Supplemental Fig. S1E of the revised manuscript. We don't have a good explanation for the discrepancy between increased urinary aldosterone and unchanged plasma renin activity in Fgf23/VDR and Klotho/VDR mutants. Because the observed increases in urinary aldosterone excretion in Fgf23/VDR and Klotho/VDR mutants were mild, it is possible that the changes in plasma renin activity were too small to be picked up by the assay. Changes in potassium homeostasis do not appear to be able to explain the observed changes in urinary aldosterone, because the changes in serum and urinary potassium were not consistent between Fgf23/VDR and Klotho/VDR mutants.

4. More generally, the authors should be more cautious when interpreting results from metabolic cages if urine output differs. Indeed, sampling of urine in metabolic cages is particularly difficult in

mouse experiments because the total amount of urine voided over 24h is very small (when compared to rats for instance) and a large amount of urine evaporates in the collecting system. Obviously the error consecutive to this evaporation is minimal when urine output is more important. Thus, the authors should consider that some differences in sodium output might be explained by urine collection differences due to differences in urinary volume. Normalizing urine parameter to urine creatinine is should address this potential problem.

Authors: It is certainly a good point that there are technical pitfalls for urine sampling in mouse experiments. However, we have many years' experience in our laboratory with urine sampling in metabolic cages in mice, and the metabolic cages we use are specifically designed to minimize evaporation of the collected urine. Therefore, it is highly unlikely that the differences in sodium output can be explained by urine collection differences. To rule out evaporation differences, we normalized all urine parameters to creatinine values in the revised manuscript, as suggested. In Figs. 3A and 6A, we also show urinary sodium excretion per 12h, because the reduced renal sodium excretion after rFGF23 treatment is caused by both reduced urinary volume and reduced urinary sodium concentration (Fig. 3A).

5. 12h hour urine sampling is also not really correct for this kind of study. Urinary sodium excretion varies along day and night. Circadian rhythm has a major effect on urinary sodium secretion. As an example, circadian rhythm might be affected in some of the knock out animals and explain the difference in natriuresis. To limit those effects, 24h urinary measurements should be performed.

Authors: In principle, we agree. The simple reason why we use a 12-hour sampling period is that our Ethical Committees and Animal Protection Law reinforce that the shortest time possible is used for metabolic cage experiments. We don't have the permission to keep the mice in metabolic cages for 24 hours in this study. We fully agree that circadian rhythm affects urinary sodium excretion. However, urine collections in our experiments were performed in all experimental groups at the same time of the day (overnight between 7 p.m. and 7 a.m.). We explicitly state this in M&M in the revised manuscript on page 17. Therefore, circadian rhythm cannot be a confounder in our study unless the different genotypes would have different circadian rhythms. It is not known and there is absolutely no indication that the mouse models used in our study are characterized by changes in circadian rhythm or behavior.

6. The reviewer is puzzled to some extent by the look of NCC band in the western blot experiments. Glycosylated NCC usually migrates as multiple bands (or smear) around 130kD and above. It doesn't seem to be the same for the NCC western blot (WB) shown here. Therefore, authors should provide full blots to reviewers and indicate molecular weight of the WB bands. In general it is not correct to show as shown here only one single or few representative blots, and it is also not correct to cut the images just around the band of interest.

Authors: The reason why the glycosylated NCC band does not look like a smear around 130 kDa in our Western blots is that we used gradient gels (4-20%). This results in a single sharp NCC band which can easily be quantified. Using homogeneous 10% gels for SDS-PAGE results in the typical smear appearance of the NCC band also in our hands (see Figure 1 below). The reason why we show only representative blots and why we cut the images around the band of interest is simply to save space in the figures. Showing entire Western blots for all figures would substantially increase the amount of space necessary for the figures. However, we included entire NCC Western blot images from all animals shown in Fig. 1C in Fig. 2 for reviewers below as suggested, and can provide full, uncropped images of all blots in source data files after conditional acceptance of the paper. In addition, we indicate the molecular weight of the Western blot bands in all revised figures.

Fig. 1. Comparison of NCC and α ENaC bands in renal cortical membrane preparations, using gradient gels (upper panels) and homogeneous 10% gels (lower panels) in combination with the same anti-NCC and anti- α ENaC antibodies.

7. *α ENaC is part of a multimeric complex forming the channel. Measurements of Beta and gamma subunit renal abundance are also necessary for interpretation. Moreover, a and g EnaC are processed as different polypeptides that should be quantified separately.*

Authors: We agree and thank the reviewer for this important comment. The reason why we focused on alpha-ENaC is that it is generally accepted that aldosterone positively regulates its expression. We included data on renal protein expression of the gamma- and beta-ENaC subunits in the revised manuscript. As shown in Supplemental Fig. S1A of the revised manuscript, renal protein expression of both gamma- and beta-ENaC subunits was lower in $Fgf23^{-}/VDR^{\Delta/\Delta}$ and $Klotho^{-}/VDR^{\Delta/\Delta}$ mutants, relative to VDR single mutants. In addition, gamma- and beta-ENaC subunits were increased in wild-type mice treated with recombinant FGF23 (Fig. 3B). These data show that alpha- versus beta- and gamma-ENaC subunits are reciprocally regulated in loss- and gain-of-Fgf23 function models. Because it is known that aldosterone up regulates alpha-ENaC (May et al., 1997), but down-regulates full length gamma-ENaC subunits (Masilamani et al., 1999), our findings suggest that this regulation may be an indirect, aldosterone-mediated process. In agreement with this notion, urinary aldosterone was higher in $Fgf23^{-}/VDR^{\Delta/\Delta}$ and $Klotho^{-}/VDR^{\Delta/\Delta}$ compound mutants vs. VDR single mutants, and lower in rFGF23-treated vs. vehicle-treated mice.

8. *The effect of Fgf23 on heart remodeling is already known as acknowledge by the authors. The hypertensive effect of rFgf23 is interesting and is well demonstrated. The authors propose that Fgf23 directly up regulates NCC thereby increasing salt retention. However, western blots test only protein abundance and do not reflect always changes in activity. The effects of fgf23 on NCC activity should be provided. This could be tested in a cell culture model. Alternatively, the changes in natriuresis induced by an acute hydrochlorothiazide (HCTZ) injection might be used to test a potential difference in NCC activity in vivo. Long-term infusion of HCTZ infusion to test NCC stimulation by fgf23 is rather indirect and doesn't reflect necessarily an activation of NCC upon fgf23 treatment. Indeed, many compensatory mechanisms can be triggered during chronic HCTZ treatment and can affect the results of the experiments presented here.*

Authors: We fully agree that data about NCC activity would strengthen the conclusions of this manuscript. Therefore, we performed intracellular sodium measurements in kidney slices from vehicle- and rFGF23 treated animals, using the sodium-sensitive molecule SBFI in combination

with 2-photon microscopy. These data are shown in Fig. 3F of the revised manuscript. rFGF23-treated mice showed an about 3-fold up regulation in fluorescence intensity in distal renal tubules, indicating that the FGF23-induced up regulation of NCC membrane expression also increased sodium entry into distal tubular epithelium. The increased cellular sodium uptake was reversed by ex vivo addition of the specific NCC inhibitor chlorothiazide to the kidney slices (Fig. 3F). To further document the direct effect of FGF23 on distal tubular sodium uptake, we treated live kidney slices from wild-type mice with FGF23 *in vitro*, and monitored intracellular sodium by 2-photon microscopy. Intracellular SFBI fluorescence in distal tubules gradually increased after addition of rFGF23, and returned to normal after addition of chlorothiazide (Fig. 3G). An additional indication of augmented NCC activity is protein phosphorylation. Therefore, we also performed Western blot analysis of phospho-NCC at positions T55, S71 and S91. Phospho-NCC expression was lower in $Fgf23^{-}/VDR^{\Delta\Delta}$ and $Klotho^{-}/VDR^{\Delta\Delta}$ compound mutants (Suppl. Fig. S1D). Conversely, rFGF23-treated mice showed an about 2-fold up regulation in phospho-NCC expression especially at S71 (Fig. 3D and Suppl. Fig. S4).

We also agree that long-term CTZ treatment can trigger compensatory mechanisms. However, we did not perform additional experiments with short-term CTZ treatment, because the intracellular sodium imaging data already provide unequivocal evidence of increased NCC function after rFGF23 treatment.

We thank the reviewer for this suggestion which has significantly strengthened the manuscript.

Minors:

It is not very clear in the legends how many animals per group have been used for biological parameters, immunofluorescence and western blot and should be provided for each type of experiments.

Authors: We have changed the manuscript accordingly. The number of independent samples is now mentioned in the figure legends for each experiment.

Reviewer 2

Andrukhova et al presents an interesting study suggesting that FGF23 modulates the expression of the renal NaCl cotransporter, NCC of the distal convoluted tubule and consequently the epithelial sodium channel, ENaC of the collecting duct. They show that renal salt reabsorption is reduced in the double knockout for FGF23 and Klotho and conversely that FGF23 administration in wild type mouse causes salt retention.

The major strength of the study is the multiple animal models in which the authors test their hypothesis and that seems to correlate the observations in opposite models. The study, however, has some important weakness that precludes the acceptance of the conclusions.

1. The study of the double knockout animals $FGF23^{-}/-$ and $VDR^{\Delta\Delta}$ or $KI^{-}/-$ and $VDR^{\Delta\Delta}$ is uncompleted as to understand the model. They show that 9 months animals exhibit a remarkable increase in salt excretion, but no other data on salt balance is presented. Are these mice dehydrated? Is blood pressure ok? If not, where do they take the excreted salt from? Are they thus eating more food? Could this be the consequence of the diet administered to these mice? If they were not eating more food, I would expect these mice to be dehydrated and hypotensive.

Authors: This point is well taken. We included blood pressure measurements by aortic catheterization in 9-month-old $Fgf23^{-}/VDR^{\Delta\Delta}$ and $Klotho^{-}/VDR^{\Delta\Delta}$ compound mutants in the revised manuscript (Fig. 2). The reviewer is right; the compound mutants demonstrate a significant decrease in systolic, diastolic and mean arterial pressure and plasma volume as compared to the

VDR^{ΔΔ} animals. Food consumption was unchanged in Fgf23^{-/-}/VDR^{ΔΔ} and Klotho^{-/-}/VDR^{ΔΔ} compound mutants, relative to VDR^{ΔΔ} and wild-type controls (Suppl. Fig. S1C). All animals were on the same diet containing 0.2% sodium. Therefore, sodium intake did not differ. Nevertheless, it is clear that sodium must be balanced in the long run. In order to adapt to reduced NCC expression and the accompanying renal sodium wasting, Fgf23/VDR and Klotho/VDR mutants up regulate aldosterone which in turn results in sodium conservation in aldosterone target organs. Therefore, we believe that sodium is balanced in Fgf23/VDR and Klotho/VDR mutants, however, at the expense of reduced blood volume.

2. If the double mutant animals discussed above has a real decrease on NCC one would expect a full Gitelman's syndrome to be developed, so it will be desirable that a complete characterization of these mice is presented. Are they hypokaliuric? Are they hypocalciuric? Do they have metabolic alkalosis? Supplemental figure 1 shows that fgf23 double mutant has lower serum K⁺ levels, but not because it is low (is above 4.0) but because it higher in the controls. In fact, in the double mutant Klotho potassium level is increased.

Authors: Also very good point. We added data about urinary potassium excretion, and urinary pH in the revised manuscript (Suppl. Fig. S1B). Urinary potassium excretion was not consistently changed in Fgf23^{-/-}/VDR^{ΔΔ} and Klotho^{-/-}/VDR^{ΔΔ} compound mutants; it was increased in Fgf23^{-/-}/VDR^{ΔΔ} mice, and decreased in Klotho^{-/-}/VDR^{ΔΔ} compound mutants, relative to wild-type controls. The reason for this difference remains unknown. Urinary pH did not differ between the genotypes (Suppl. Fig. S1B). Furthermore, Fgf23^{-/-}/VDR^{ΔΔ} and Klotho^{-/-}/VDR^{ΔΔ} compound mutants are hypercalciuric (Andrukhova et al., EMBO J, in press). Therefore, although Fgf23^{-/-}/VDR^{ΔΔ} and Klotho^{-/-}/VDR^{ΔΔ} compound mutants are characterized by reduced renal NCC expression and increased aldosterone, they do not develop a typical Gitelman's syndrome. Rather, the phenotype is more complex, because Fgf23 signaling targets SGK1 and WNK4 in distal tubular epithelium. SGK1 and WNK4 are central regulators of ion channel trafficking in distal tubules, and regulate not only NCC but also other ion channels such as TRPV5 and ROMK1. At present, we are only at the beginning of a thorough understanding of all the ramifications of FGF23 signaling in distal tubular epithelium. More details may come to the surface in the near future. We added a paragraph in the discussion of the revised manuscript (page 13) where we address this interesting point.

3. The antibodies used for NCC and WNK4 are commercially developed and poorly characterized. Most studies using good antibodies for NCC shows a smear band corresponding to glycosylated form of the protein. In this study only a fine band is presenting. In addition, this reviewer believes that in this type of studies it is more convincing when the data from all animals is presented. If figure 1C or 1D was done using between 8 and 12 animals, why not showing all of them to the reader can judge for itself the results. Also, since authors are using a commercial antibody not well characterized, it will be better if they show the entire blot, so the reader can judge the size of the bands that come out with the antibodies. In the present form we have no clue of what size the presented bands are. This is also valid for ENaC, which usually is presented as a double band, so we do not know which band is the one that is presented.

Authors: The reason why the glycosylated NCC band does not look like a smear in our Western blots is that we used gradient gels (4-20%). This results in a single sharp NCC band which can easily be quantified. Using homogeneous 10% gels for SDS-PAGE results in the typical smear appearance of the NCC band and a double band for ENaC also in our hands (see Figure 1 above). We added size markers to all Western blot images. To save space, we show only representative blots in the figures in the manuscript. In the Fig. 2 for reviewers (see below) we provide entire blots for NCC used for the Figure 1C in the manuscript. The data from all 9-month-old WT, VDR^{ΔΔ} and Fgf23^{-/-}/VDR^{ΔΔ} animals is presented. It is clearly evident that NCC is down regulated in Fgf23^{-/-}/VDR^{ΔΔ} mice.

Fig. 2. Entire Western blot images of NCC protein expression in renal cortical membrane preparations of 9-month-old wild-type, VDR single mutant, and *Fgf23*/VDR compound mutant mice.

4. The antibody for *WNK4* is also a commercial one that has not been properly studied. In the *WNK* field it is well known that one problem has been the lack of a truly good anti-*WNK4* antibody detecting *WNK4* in renal tissue.

Authors: We characterized the anti-*WNK4* antibody used in our study in more detail recently (Andrukhova et al., EMBO J, in press). Although it is certainly not perfect, this antibody is working well in immunoblots and immunohistochemistry. The polyclonal anti-*WNK4* antibody used by us recognizes one major band at approximately 135 kDa, and two probably nonspecific bands at 70 and 20 kDa in Western blots. We present an entire Western blot image for *WNK4* protein expression in

wild-type mice treated with recombinant FGF23 or vehicle in Fig. 3 for reviewers shown below, and we can provide source data files upon conditional acceptance of the paper.

Fig. 3. Entire Western blot image of WNK4 protein expression in total kidney homogenates of wild-type mice treated with recombinant FGF23 or vehicle for 5 days.

5. The effect of rFGF23 on NCC is not necessary due to direct effect on the cotransporter in DCT. FGF23 could have an effect on other systems, like adrenergic or renin-angiotensin that is known to activate NCC. The experiment with dissected tubules could help in this issue. However, isolated number of DCTs isolated this way is very limited. I suggest doing the following. On the one hand to get DCTs isolated by the COPAS methodology that has been recently mastered by Bindel's group in Netherlands or by Loffing in Zurich, this way the amount of DCTs for study are seriously increased. On the other hand, to repeat the DCT experiment shown in Fig 2F but dissecting the tubules from the FGG23 knockout mice.

Authors: We agree that the effect of FGF23 on NCC could be indirect. This is exactly the reason why we performed the experiment in isolated distal tubules, namely to show the direct effect of FGF23 signaling on distal tubular epithelium. The procedure for manual isolation of renal distal tubular segments is well established in our lab and is working well in our hands. However, as correctly mentioned by the Reviewer this type of isolation is time consuming and needs experience. To rule out contamination with proximal tubules, we performed purity and quality controls, using mRNA expression of distal (TRPV5, calbindin 28k) and proximal (NaPi-2a, NaPi-2c) tubule specific genes. We added this information to M&M of the revised manuscript. Because the method is working very well in our hands, we did not change the isolation method.

According to the Reviewer's suggestion we included an experiment with isolated renal distal tubular segments from *Fgf23*^{-/-} mice in the revised Fig. 5A. Similar to distal tubular segments isolated from wild-type mice, we observed a marked increase in NCC protein expression in recombinant FGF23 (rFGF23)-treated segments from *Fgf23*^{-/-} mice, an effect which was abrogated by co-treatment with ERK1/2 and SGK1 inhibitors (Fig. 5A). To further demonstrate the direct activation of NCC by FGF23 signaling, we quantified phospho-NCC in isolated distal tubular segments from wild-type and *Klotho* deficient mice after treatment with vehicle and rFGF23 (Fig. 5B). rFGF23 increased phosphorylation of NCC in isolated distal tubular segments in a *Klotho* dependent manner, showing that the effect of FGF23 on distal tubular cells is direct. We thank the reviewer for this suggestion which has significantly strengthened the manuscript.

6. Because of this previous comment, the experiments with chlorithiazide does not help because we cannot be sure that rFGF23 is actually activating NCC or activating something else that in turn activates NCC, so thiazide would prevents that. In fact, it is surprising that with only five days of rFGF23 administration is enough to cause even cardiac hypertrophy due to hypertension that is secondary to increased salt reabsorption by NCC.

Authors: It is true that it is surprising that only 5 days of rFGF23 administration already caused heart hypertrophy in a Klotho dependent fashion. We were also surprised. Obviously, rFGF23 has very strong effects on volume regulation. To show that rFGF23 is directly activating NCC, we performed additional experiments in isolated distal tubular segments. See our answer to comment #5.

7. How the authors conceal their observation that inhibiting ERK1/2 prevents the increased expression of NCC by FGF23 with published evidences that activation of ERK1/2 is associated with decreased expression of NCC (1, 2), so inhibition of ERK1/2 would be expected to be associated with up-regulation of NCC.

1. Ko B, Joshi LM, Cooke LL, Vazquez N, Musch MW, Hebert SC, Gamba G, and Hoover RS. Phorbol ester stimulation of RasGRP1 regulates the sodium-chloride cotransporter by a PKC-independent pathway. Proc Natl Acad Sci USA 104: 20120-20125, 2007.

2. Ko B, Kamsteeg EJ, Cooke LL, Moddes LN, Deen PM, and Hoover RS. RasGRP1 stimulation enhances ubiquitination and endocytosis of the sodium-chloride cotransporter. Am J Physiol Renal Physiol 299: F300-F309, 2010.

Authors: We cannot entirely explain the discrepancies between our study and those of Ko et al. in the regulation of NCC by ERK1/2 activation. It is correct that Ko et al. reported that ERK1/2 signaling is involved in NCC ubiquitination and endocytosis through possible activation of Ras/GRP1, Ras, Raf and MEK1/2. On the other hand, we recently established the FGFR/Klotho-ERK1/2-SGK1-WNK4 signaling pathway in distal tubular cells (Andrukhova et al., EMBO J, in press). Isolated tubular segments are probably the best in vitro system to test the cellular response to certain molecules because the cells are polarized and remain in their natural environment. Using this system, ERK1/2 and SGK1 inhibition prevented the rFGF23-associated up regulation in NCC expression in our experiments. Therefore, we are convinced that activation of ERK1/2 is necessary for downstream activation of SGK1 and WNK4 to up regulate NCC membrane expression. However, ERK1/2 is involved in multiple signaling pathways. Therefore, it is possible that ERK1/2 is also involved in signaling pathways leading to NCC degradation, depending on the upstream activator and its specific downstream activation pattern. Clearly, further experimentation is required to fully elucidate the FGF23 signaling pathway and its interaction with other pathways in distal tubular epithelium. We added a statement about this in the discussion of the revised manuscript on page 15.

Reviewer 3

The present paper describes the undescribed role of FGF23 of regulating Na⁺ homeostasis, systemic blood pressure and heart hypertrophy. By using compound mutant mice lacking Fgf23 or klotho and VDR, the Authors found that double mutant ko mice showed renal Na⁺ wasting with respect to VDR mutants and WT animals, associated with elevated urinary aldosterone, decreased NCC but up regulated ENaC. In contrast, infusion of recombinant FGF23 in control mice induced Na⁺ retention and volume expansion which possibly increased blood pressure and induced heart hypertrophy. Mechanisms underlying the effect of FGF23 on Na⁺ retention and its cardiovascular consequences were explored in vitro and in vivo. The effect was attributed to FGF23-induced up regulation of distal renal tubular NCC.

The paper addresses a novel issue. The experimental design is well done, the methodological approach is sophisticated and the conclusions are in general supported by data. Few additional experiments could help strengthen the conclusions. The following is for the Authors' consideration:

1. The Authors showed that cardiovascular effects of increased circulating FGF23 are mediated through the up regulation of distal renal tubular NCC by studies employing the NCC functional blocker chlorothiazide. To finally demonstrate specificity of the action of Na⁺:Cl⁻ cotransporter, NCC ko animals should be used.

Authors: In principle, we agree with this suggestion. However, it is well known that NCC knockout animals develop a phenotype similar to Gitelman's syndrome with significantly elevated plasma aldosterone levels, hypocalciuria, hypomagnesemia, and compensated alkalosis. Moreover, loss of NCC leads to major structural remodeling of the renal distal tubule that goes along with marked changes in glomerular and tubular function. Therefore, treatment effects of FGF23 might be masked by the complex phenotype of NCC knockout animals. Therefore, we decided to use an alternative approach: i) We performed intracellular sodium measurements in kidney slices from vehicle- and rFGF23 treated animals, using the sodium-sensitive molecule SBFI in combination with 2-photon microscopy. These data are shown in Fig. 3F of the revised manuscript. rFGF23-treated mice showed an about 3-fold up regulation in fluorescence intensity in distal renal tubules, indicating that the FGF23-induced up regulation of NCC membrane expression also increased sodium entry into distal tubular epithelium. The increased cellular sodium uptake was abrogated by ex vivo addition of the specific NCC inhibitor chlorothiazide to the kidney slices (Fig. 3F). To further document the direct effect of FGF23 on distal tubular sodium uptake, we treated live kidney slices from wild-type mice with FGF23 *in vitro*, and monitored intracellular sodium by 2-photon microscopy. Intracellular SBFI fluorescence in distal tubules gradually increased after addition of rFGF23, and returned to normal after addition of chlorothiazide (Fig. 3G). ii) An additional indication of augmented NCC activity is protein phosphorylation. Therefore, we also performed Western blot analysis of phospho-NCC at positions T55, S71 and T91. Fgf23^{-/-}/VDR^{ΔΔ} and Klotho^{-/-}/VDR^{ΔΔ} compound mutants demonstrated lower phospho-NCC expression (Suppl. Fig. S1D). Conversely, rFGF23-treated mice showed an about 2-fold up regulation in phospho-NCC expression (Fig. 3D). We feel that these additional data provide convincing evidence of the specificity of the FGF23-driven NCC regulation.

2. Given the key role of Na⁺ excretion in the described experiments, data on Na⁺ urinary excretion should be expressed as Ur Na/Crea (mmol/mmol) in all the studies (Figure 2A, 2E, 3A) to avoid any bias that reduced Na⁺ excretion could only reflect urine volume reduction after rFGF23 injection.

Authors: According to the reviewer's suggestion we present Na⁺ urinary excretion data as Ur Na/Crea (mmol/mmol) in the revised manuscript. In addition, we show Na⁺ excretion per 12 hours in Figs. 3A and 6A because the Na⁺ retention after rFGF23 injection is a combination of reduced urinary output and reduced urinary Na⁺ concentration (Fig. 3A).

3. Emerging evidence indicates the relationship between Vitamin D, FGF23 and RAAS as nicely summarized by de Borst et al (JAm Soc Nephrol 22:1603, 2011). In conditions of high FGF23 levels, reduced activation of vitamin D up regulates renal renin production. To rule out that the cardiovascular effects of rFGF23 injection could be attributable to RAAS activation, plasma renin activity should be assessed in mice injected with rFGF23.

Authors: This is also a good point. FGF23 could indirectly stimulate RAAS activity by suppressing vitamin D hormone production. Therefore, we measured plasma renin activity in loss- and gain-of-function experiments. As shown in Supplemental Fig. S1E of the revised manuscript, plasma renin activity in Fgf23^{-/-}/VDR^{ΔΔ} and Klotho^{-/-}/VDR^{ΔΔ} mutants was similar to that of VDR^{ΔΔ} animals. Moreover, plasma renin activity remained unchanged (Fig. 3A) and serum aldosterone was lower (Fig. 3A) in animals treated with recombinant FGF23 for 5 days. These results suggest that the cardiovascular effects of FGF23 are RAAS independent. In addition, it is of note that the up regulated renin expression in VDR deficient mice reported earlier (Li et al., JCI 2002) is an indirect effect caused by severe secondary hyperparathyroidism (Andrukhova et al., Mol Endocrinol, in press), and is not seen in normocalcemic VDR mutant mice on rescue diet (Suppl. Fig. S1E).

4. *The present experiments have been performed with animals fed a normal Na⁺ diet. It should be interesting to study the effect of the injection of rFGF23 in animals fed a low or high Na⁺ diet.*

Authors: We thank the reviewer for this excellent suggestion. We performed an additional experiment in vehicle- and rFGF23-treated mice kept on normal, low and high sodium diet. The data are shown in Fig. 7 of the revised manuscript. Treatment with recombinant FGF23 increased mean arterial pressure (MAP) in mice on all three diets (Fig. 7A). However, intriguingly we found an inverse relationship between dietary sodium and the increase in MAP caused by rFGF23 treatment (Fig. 7A). Treatment with rFGF23 suppressed urinary aldosterone in mice on all diets, but the remaining aldosterone excretion was inversely related to dietary sodium (Fig. 7C). The higher the remaining urinary secretion of aldosterone, the stronger the effect of rFGF23 on NCC, sodium retention, and blood pressure was (Fig. 7A-D). Thus, aldosterone is obviously a very important modulator of the hypertensive effect of rFGF23. We currently believe that aldosterone might augment rFGF23 signaling at the level of SGK1 because both signaling pathways converge on SGK1 (Fig. 7E). This finding may be clinically very relevant, because CKD patients show an up regulation of both the FGF23 and the aldosterone signaling axis. We added a paragraph in the discussion of the revised manuscript on page 14-15 to explain this.

5. *Serum Na⁺ levels are unchanged in double FGF23 mutant mice but not in double klotho mutant mice (Figure S1). The Authors should explain why.*

Authors: It is true that serum sodium levels were reduced in Klotho/VDR but not in Fgf23/VDR mutants, relative to wild-type. However, we don't think that this is a biologically important difference, because there were no significant differences between Klotho/VDR and Fgf23/VDR vs. VDR mutants in both experiments.

6. *WB image of NCC in VDRdelta/delta mice should be replaced (Figure 1D).*

Authors: We replaced the Western blot image shown as suggested.

7. *Serum Na⁺ levels are significantly increased in rFGF23 injected animals vs vehicle but the mean values seem comparable.*

Authors: We split the y-axis in Fig. 3A of the revised manuscript to better show the differences. It is now clearly evident that serum Na⁺ was higher in rFGF23-injected than in vehicle-treated mice.

8. *Basal values of urinary Na⁺ excretion vary among the experiments (Figure 2A, 2E and supplemental Figure 2, Figure 3A). Same variability is present for heart/body weight values in vehicle treated animals (Figure 2D and Supplemental Figure 3). The Authors should explain such a variability.*

Authors: The reasons for the different values for urinary Na⁺ excretion are i) different age of the animals, and ii) different diets used in each particular experiment. Mice in Figs. 2A, 2E, and 3A were of the same age (3 months), but the mice in Fig. 2A (now 3A) and 3A (now 6A) were on normal diet, whereas the mice in Fig. 2E (now 4B) were on rescue diet. Although the Na⁺ content of both diets is similar (0.2%), the different diets may account for the observed differences in Na⁺ excretion. In experiments involving single Fgf23^{-/-} or Klotho^{-/-} knockout animals, 4-week-old animals on rescue diet were used (Suppl. Fig. S2), whereas data from 9-month-old mice on rescue diet are shown in Fig. 1A.

The same is true for the heart/body weight values. In Fig. 2D (now 4A), 3-month-old animals, and in Supplemental Fig. S3 (now S5), 4-week-old animals were used. We included this information in the legends of the revised figures.

2nd Editorial Decision

22 December 2013

Thank you for the re-submission of your manuscript to EMBO Molecular Medicine. We have now heard back from the three Reviewers whom we asked to evaluate your manuscript. You will see that all three Reviewers are now generally supportive of your work although a few concerns are expressed, in particular by Reviewer 2, that prevent us from considering publication at this time.

Reviewer 1 would like you to introduce a clarifying statement with respect to the fact that animals with ablation of fgf23 have elevated natriuresis.

Reviewer 2 comments on the provided Figure 2 and remains unconvinced of the specificity of the anti- NCC and -WNK4 antibodies, and suggests that blots should be presented showing kidneys from NCC and WNK4 knockout mouse to prove antibody specificity. Please note that, in case your EMBO Journal paper should still not be publicly available at the time of submission of your revision, the manuscript should be made available. Reviewer 2 is also concerned that the methodology used to measure blood pressure is sub-optimal and challenges the appropriateness of waiting 9 months of age to carry out such assessments

Reviewer 3 acknowledges that Na⁺ urinary excretion is now calculated as Ur Na/Crea but notes that reduction after rFGF23 administration does not reach statistical significance and suggests increasing the number of experimental points. S/he would also like you to better discuss the observation that a low Na⁺ diet exacerbated the FGF23-induced increase in MAP and inversely correlated with aldosterone excretion.

Considered all the above, while publication of the paper cannot be considered at this stage, we would be prepared to consider a suitably revised submission, with the understanding that the Reviewers' concerns must be fully addressed with additional experimental data where appropriate and that acceptance of the manuscript will entail a second round of review.

Please note that it is EMBO Molecular Medicine policy to allow a single round of revision only and that, therefore, acceptance or rejection of the manuscript will depend on the completeness of your responses included in the next, final version of the manuscript.

As you know, EMBO Molecular Medicine has a "scooping protection" policy, whereby similar findings that are published by others during review or revision are not a criterion for rejection. However, I do ask you to get in touch with us after three months if you have not completed your revision, to update us on the status. Please also contact us as soon as possible if similar work is published elsewhere.

I look forward to seeing a revised form of your manuscript as soon as possible.

***** Reviewer's comments *****

Referee #1 (Comments on Novelty/Model System):

The study deciphers some of the mechanisms that link high FGF23 levels to hypertension. High Fgf23 levels are seen in patients with chronic renal failure. Since these patients often exhibit hypertension and cardiac hypertrophy the observation of the authors is of major interest and is expected to open avenues for new therapeutic strategies

Referee #1 (Remarks):

The authors have adequately and carefully addressed my initial concerns. The only point that remains unclear to me is their explanation of the observation the animals with ablation of *fgf23* have elevated natriuresis. It is indeed impossible to state that they are in balance if the intake are identical, even if they have low plasma volume. The authors have measured the food intake and it was not different between groups. However, this does not imply that absorption of sodium by the gut is the same. This should be clearly acknowledged.

Referee #2 (Comments on Novelty/Model System):

Most of my previous concerns were resolved, but there is still some issues left that I still found difficult to accept.

I am still not convinced with the specificity of the antibodies used for NCC and WNK4. Note that in figure 2 provided by the authors in the answer to reviewers file, there are two lines in the upper left blot and two in the upper right blot, unmarked but showing exactly the same band. I cannot see the Andrukhova et al., EMBO J, in press since it has not been released on line. There are knockout mice for NCC and for WNK4 in the literature, so it will be of help if authors present a western blot in which kidney from NCC and WNK4 knockout mice is present to demonstrate the specificity of the antibodies.

Why the authors choose the aortic catheterization of nine-month-old mice better and more precise methodology exist to assess blood pressure. Measurement in the awaked animal is always better. Ideally should be done with radiotelemetry, but in case it not an accessible technology tail cuff pletismography can be of help. In addition, if the NCC defect is suppose to be there since bird due to the knockout of *fgf23*, why waiting until nine months when BP can be assessed at two months of age or so.

Referee #2 (Remarks):

The manuscript improved most concerns were either resolved or explained better. I still find, however, that problems with key observations were not resolved. For instance, a better work can be done to be sure that antibodies are specific for the target protein. Also, assessing blood pressure in the anesthetized animal is not the ideal way to measure blood pressure.

Referee #3 (Remarks):

In general the manuscript has been improved by the Authors who provided additional data in support of the impact of FGF23 on renal sodium handling. The Authors have made additional experiments that addressed most of my and other Reviewers' concerns on the previous manuscript.

The expression of Na⁺ urinary excretion calculated (as suggested) as Ur Na/Crea showed only a trend toward reduction after FGF23 which however, did not reach statistical significance. The Authors should improve these data by studying additional animals.

The new experiments with different Na⁺ diets have given surprising results, as correctly pointed out by the Authors. However, that a low Na⁺ diet worsened the FGF23-induced increase in MAP and inversely correlated with aldosterone excretion merits further explanations in the discussion.

Some inconsistencies are present along the text.

Figure 1A and Figure 7A: serum and urinary Na⁺ should be reported as written in the results and not the opposite.

Text referring to Figure S1D and Figure 2 should report FGF23 $-/-$ /VDR / mice before Kl $-/-$ /VDR / .

Page 9, 2 photon experiments are wrongly cited in the text: Figure 3E should be 3F.

1st Revision - authors' response

06 February 2014

We would like to thank the reviewers for the constructive critique.

Reviewer 1

The authors have adequately and carefully addressed my initial concerns. The only point that remains unclear to me is their explanation of the observation the animals with ablation of fgf23 have elevated natriuresis. It is indeed impossible to state that they are in balance if the intake are identical, even if they have low plasma volume. The authors have measured the food intake and it was not different between groups. However, this does not imply that absorption of sodium by the gut is the same. This should be clearly acknowledged.

Authors: We fully agree. It is clear that the counter-regulatory increase in circulating aldosterone in Fgf23/VDR and Klotho/VDR compound mutants will increase intestinal Na absorption despite similar food intake. Actually, this is what we meant with our statement on page 7 that aldosterone conserves Na⁺ in aldosterone target organs. We added a sentence to clearly spell this out in the revised manuscript (last para, page 7).

Referee 2

Most of my previous concerns were resolved, but there is still some issues left that I still found difficult to accept.

I am still not convinced with the specificity of the antibodies used for NCC and WNK4. Note that in figure 2 provided by the authors in the answer to reviewers file, there are two lines in the upper left blot and two in the upper right blot, unmarked but showing exactly the same band. I cannot see the Andrukhova et al., EMBO J, in press since it has not been released on line.

Authors: The band shown in the unmarked lanes is also NCC. The reason why we did not mark the lanes is that the data were not included in Figure 1C. The figure for reviewers exemplarily shows the entire Western blot images of the data depicted in Figure 1C. We labeled all lanes now (Fig. 1 for reviewers, see below). The two lanes in the upper left blot and the first lane in the upper right blot are kidney samples from Klotho^{-/-}/VDR^{ΔΔ} mutants. The last samples in the upper left and upper right blots are test samples from kidneys of wild type animals not involved in this study and used as positive controls. Furthermore, we confirmed the specificity of the anti-NCC antibody by using kidney extracts from NCC knockout mice (see below).

Fig. 1. Entire Western blot images of NCC protein expression in renal cortical membrane preparations of 9-month-old wild-type, VDR single mutant, and *Fgf23*/VDR or *Klotho*/VDR compound mutant mice.

2. There are knockout mice for NCC and for WNK4 in the literature, so it will be of help if authors present a western blot in which kidney from NCC and WNK4 knockout mice is present to demonstrate the specificity of the antibodies.

Authors: This is a very good suggestion which has further strengthened our study. We tested the anti-NCC and anti-WNK4 antibodies by using total protein extracts from kidneys of NCC (generous gift of Dr. Manoocher Soleimani, Cincinnati University, OH, USA) and of WNK4 (generous gift of Dr. Gerardo Gamba, Molecular Physiology Unit, Mexico City, Mexico) knockout mice. Both

antibodies did not give a signal in kidney protein samples isolated from the corresponding knockout mice, confirming their specificity. The results are shown in Supplementary Fig. S5.

3. Why the authors choose the aortic catheterization of nine-month-old mice better and more precise methodology exist to assess blood pressure. Measurement in the awaked animal is always better. Ideally should be done with radiotelemetry, but in case it not an accessible technology tail cuff pletismogarchy can be of help. In addition, if the NCC defect is suppose to be there since bird due to the knockout of fgf23, why waiting until nine months when BP can be assessed at two months of age or so.

Authors: We agree that radiotelemetry is a very useful method to monitor blood pressure in awake animals. Unfortunately, we do not have access to this technology in our laboratory. We performed central arterial blood pressure measurement in our study, because it is the gold standard for blood pressure measurements in mice. The noninvasive tail cuff plethysmography has the advantage of making blood pressure measurements possible in non-anesthetized animals, but it is well known that this method is imprecise and not very reliable in mice (Plehm R, Barbosa ME, Bader M. Animal models for hypertension/blood pressure recording. *Methods Mol Med.* 2006;129:115-26). Therefore, we chose to employ central arterial blood pressure measurement in our study to be on the safe side. It is clear that the blood pressure measurements can also be done in younger mice. The reason why we used 9-month-old mice for the blood pressure measurements shown in Figure 2 is simply that the mineral homeostasis data shown in Figure 1 are all from 9-month-old mice. We thought that it is most informative to use age-matched mice for mineral homeostasis and blood pressure measurements in Figures 1 and 2.

Referee #3

In general the manuscript has been improved by the Authors who provided additional data in support of the impact of FGF23 on renal sodium handling. The Authors have made additional experiments that addressed most of my and other Reviewers' concerns on the previous manuscript.

1. The expression of Na⁺ urinary excretion calculated (as suggested) as Ur Na/Crea showed only a trend toward reduction after FGF23 which however, did not reach statistical significance. The Authors should improve these data by studying additional animals.

Authors: According to the reviewer's suggestion we performed additional experiments, and increased the animal numbers per group for measurement of serum and urinary Na concentration in the revised Fig. 3A (now n = 15 – 17). Ur Na/Crea is now significantly decreased in FGF23-treated mice, relative to vehicle-treated controls.

2. The new experiments with different Na⁺ diets have given surprising results, as correctly pointed out by the Authors. However, that a low Na⁺ diet worsened the FGF23-induced increase in MAP and inversely correlated with aldosterone excretion merits further explanations in the discussion.

Authors: We agree that our finding that a low Na⁺ diet aggravated the hypertensive effect of FGF23 is indeed surprising. We explained this in more detail in the discussion of the revised manuscript (page 15). Our current hypothesis is that both FGF23 and aldosterone converge on SGK1. However, it is clear that more work needs to be done to nail down the exact molecular interactions between aldosterone and FGF23 signaling.

Some inconsistencies are present along the text.

3. Figure 1A and Figure 7A: serum and urinary Na⁺ should be reported as written in the results and not the opposite.

Authors: We switched the panels showing serum and urinary Na⁺ in Figure 7B. Figure 1A does not contain data on serum Na⁺.

4. Text referring to Figure 1D and Figure 2 should report FGF23 -/-/VDR Δ/Δ mice before Kl-/-/VDR Δ/Δ.

Authors: We changed the text accordingly.

Page 9, 2 photon experiments are wrongly cited in the text: Figure 3E should be 3F.

Authors: We apologize for this mistake, and have amended the text accordingly.

3rd Editorial Decision

14 February 2014

Thank you for the submission of your revised manuscript to EMBO Molecular Medicine. We have now received the enclosed reports from the Reviewers that were asked to re-assess it. As you will see the reviewers are now supportive and I am pleased to inform you that we will be able to accept your manuscript pending the following final issues:

1) While performing our pre-publishing quality control and image screening routines, we noticed an issue pertaining to Fig. 5B, specifically there appears to be a discontinuity between the first and second lanes in the blot. I must therefore ask you to please provide us with the full source data set for this figure and an explanation.

2) Independently from item 1 above, we are now encouraging the publication of source data, particularly for electrophoretic gels and blots, with the aim of making primary data more accessible and transparent to the reader. Would you be willing to provide a PDF file per figure that contains the original, uncropped and unprocessed scans of all or at least the key gels used in the manuscript? The PDF files should be labeled with the appropriate figure/panel number, and should have molecular weight markers; further annotation may be useful but is not essential. The PDF files will be published online with the article as supplementary "Source Data" files. If you have any questions regarding this just contact me.

3) We would need a short list (up to 5) of bullet points that summarize the key NEW findings. The bullet points should be designed to be complementary to the abstract and will be used online in our new online platform.

4) As per our Author Guidelines, the description of all reported data that includes statistical testing must state the name of the statistical test used to generate error bars and P values, the number (n) of independent experiments underlying each data point (not replicate measures of one sample), and the actual P value for each test (not merely 'significant' or 'P < 0.05').

Please submit your revised manuscript within two weeks. I look forward to seeing a revised form of

your manuscript as soon as possible.

***** Reviewer's comments *****

Referee #2 (Remarks):

Agree with changes

Referee #3 (Remarks):

The Authors have adequately addressed my concerns. The manuscript is now acceptable for publication.

2nd Revision - authors' response

27 February 2014

I uploaded the revised manuscript this afternoon. However, it has not converted into a merged pdf yet. I am going to check tomorrow again.

We replaced Figs. 1, 3, and 5. Fig. 1 because the blue bars in Fig. 1C had slightly different hues. Fig. 3 because the labels in the right panel of Fig. 3E were missing. In Fig. 5, we replaced the Western blot image in the left upper panel in Fig. 5B. The explanation for the discontinuity between the first and second lanes is that we had straightened the band in the first lane. We have replaced this image by the same blot with the unprocessed image.

I uploaded source data files for all Western blot images. Some Western blot images in the figures are compressed or stretched to equalize the size in complex figures. To reflect this, we tried to match the sizes of the frames in the source data with the image shown in the figures. In addition, for Fig. 5B, we provided source data from all blots used in the bar graphs.

In the figure legends, we now indicate the statistical test used and exact p values wherever possible. We also changed the legends of the supplementary figures to indicate the statistical tests. We left the thresholds, for example $p < 0.05$, in instances where the symbols stand for many individual p values (e.g., 8 different individual p values for * and # in Fig. 1B), and where individual p values would therefore make the legend very hard to read. I hope that this is acceptable. Changes are marked in yellow.

We are suggesting the following bullet points:

- * FGF23 regulates membrane abundance and activity of the renal sodium chloride channel NCC through the ERK1/2 - SGK1 - WNK4 signaling pathway.
- * FGF23 is a sodium-conserving hormone.
- * Elevated circulating FGF23 leads to volume expansion, hypertension and cardiac hypertrophy in a Klotho-dependent fashion.
- * The NCC inhibitor chlorothiazide blunts the FGF23-induced hypertension.
- * A low sodium diet aggravates the hypertensive action of FGF23 through crosstalk with aldosterone signaling at the level of SGK1.